# Linear Causal Representation Learning from Unknown Multi-node Interventions

**Burak Varıcı**[*]
Carnegie Mellon University

**Emre Acartürk**
Rensselaer Polytechnic Institute

**Karthikeyan Shanmugam**
Google DeepMind

**Ali Tajer**
Rensselaer Polytechnic Institute

## Abstract

Despite the multifaceted recent advances in interventional causal representation learning (CRL), they primarily focus on the stylized assumption of single-node interventions. This assumption is not valid in a wide range of applications, and generally, the subset of nodes intervened in an interventional environment is *fully unknown*. This paper focuses on interventional CRL under unknown multi-node (UMN) interventional environments and establishes the first identifiability results for *general* latent causal models (parametric or nonparametric) under stochastic interventions (soft or hard) and linear transformation from the latent to observed space. Specifically, it is established that given sufficiently diverse interventional environments, (i) identifiability *up to ancestors* is possible using only *soft* interventions, and (ii) *perfect* identifiability is possible using *hard* interventions. Remarkably, these guarantees match the best-known results for more restrictive single-node interventions. Furthermore, CRL algorithms are also provided that achieve the identifiability guarantees. A central step in designing these algorithms is establishing the relationships between UMN interventional CRL and score functions associated with the statistical models of different interventional environments. Establishing these relationships also serves as constructive proof of the identifiability guarantees.

## 1 Introduction

Causal representation learning (CRL) is a major leap in causal inference, moving away from the conventional objective of discovering causal relationships among a set of variables and learning the variables themselves. By combining the strengths of causal inference and machine learning, CRL specifies data representations that facilitate reasoning and planning [1]. CRL is motivated by the premise that in a wide range of applications, a lower-dimensional latent set of variables with causal interactions generates the usually high-dimensional observed data. Therefore, CRL's objective is to use the observed data and learn the latent causal generative factors, which include the causal latent variables and their causal relationships.

**CRL objectives.** Formally, consider a set of latent causal random variables $Z \in \mathbb{R}^n$ and a directed acyclic graph (DAG) $\mathcal{G}$ that encodes the causal relationships among $Z$. The latent variables are transformed by an *unknown* function $g$ to generate the *observed* random variables $X \in \mathbb{R}^d$, where $X \triangleq g(Z)$. CRL aims to use $X$ to recover the latent causal variables $Z$ and the causal structure $\mathcal{G}$.

Two central questions of CRL pertain to *identifiability*, which refers to determining the conditions under which $Z$ and $\mathcal{G}$ can be recovered, and *achievability*, which refers to designing CRL algorithms that can achieve the foreseen identifiability guarantees. Identifiability has been demonstrated to

---

[*]Work was done while BV was a Ph.D. student at Rensselaer Polytechnic Institute.

38th Conference on Neural Information Processing Systems (NeurIPS 2024).

be inherently under-constrained [2], prompting the development of diverse methodologies that incorporate inductive biases to enable identifiability. *Interventional* CRL is one direction with significant recent advances in which interventions on latent causal variables are used to create statistical diversity in the observed data [1, 3–6].

**Unknown multi-node interventions.** Despite covering many aspects of interventional CRL, such as parametric versus nonparametric causal models, parametric versus nonparametric transformations, and intervention types, the majority of the existing studies assume that the interventions are single-node, i.e., exactly one latent variable is intervened in each environment [3–10]. This assumption, however, is restrictive in some of the application domains of CRL such as biology and robotics in which generally the subset of nodes intervened in an intervention environment can be *fully unknown*. For instance, biological perturbations in genomics are imperfect interventions with off-target effects on other genes [11, 12], or interventions on robotics applications are likely to affect multiple causal variables [13]. Hence, realizing the promises of CRL critically hinges on dispensing with the assumption of single-node interventions.

In this paper, we address the open problem of using *unknown multi-node (UMN) stochastic* interventions to recover the latent causal variables $Z$ and their causal graph $\mathcal{G}$, wherein each environment an unknown subset of nodes are intervened. We consider a general latent causal model (parametric or nonparametric) and focus on the *linear* transformations as an important class of parametric transformation models. We establish identifiability results and design algorithms to achieve them under both soft and hard interventions. For this purpose, we delineate connections between UMN interventions and the properties of score functions, i.e., the gradients of the logarithm of density functions. This score-based framework is the UMN counterpart of the single-node framework proposed in [5, 7], albeit with significant technical differences. Our contributions are summarized below.

- We show that under sufficiently diverse interventional environments, UMN stochastic hard interventions suffice to guarantee perfect identifiability of the latent causal graph and the latent variables (up to permutations and element-wise scaling).
- We show that under sufficiently diverse interventional environments, UMN soft interventions guarantee identifiability up to ancestors – transitive closure of the latent DAG is recovered, and latent variables are recovered up to a linear function of their ancestors. Remarkably, these guarantees match the best possible results in the literature of single-node interventions.
- We design score-based CRL algorithms for implementing CRL with UMN interventions with provable guarantees. These guarantees also serve as constructive proof steps of the identifiability results.

**Challenges of UMN interventions.** There are two broad challenges specific to addressing the UMN intervention setting that render it substantially distinct from the single-node (SN) intervention setting. First, in SN interventions, since the learner knows exactly one node is intervened in each environment, it can readily identify the intervention targets up to a permutation. In contrast, in UMN interventions, the learner does not know how many nodes are intervened in each environment. Therefore, the nature of resolving the uncertainty about the intervention targets becomes fundamentally different. An immediate impact of this is that it becomes more challenging to properly capitalize on the statistical diversity embedded in the interventional data. Secondly, in SN interventions, only one causal mechanism changes across the environments. Such sparse variations of the causal mechanisms are a core property leveraged by various existing CRL approaches, e.g., contrastive learning [8], and score-based framework [6, 7]. On the contrary, UMN interventions allow for many concurrent causal mechanism changes, which renders leveraging sparsity patterns in mechanism variations futile. Finally, since intervention targets are unknown, our central algorithmic idea is to properly aggregate the UMN interventional environments to create new distinct environments under which the inherent statistical diversity is more accessible.

## 1.1 Related literature

**Single-node interventional CRL.** The majority of the studies on interventional CRL focus on SN interventions [3–10, 14], which can be categorized based on their assumptions on the latent causal model, transformation, and intervention model. Based on this taxonomy, it has been shown that SN hard interventions suffice for identifiability with general latent causal models and linear transformations (one intervention per node) [7], with linear Gaussian latent models and general transformations (one intervention per node) [8], and with general latent models and general transformations (two interventions per node) [6, 9]. For the less restrictive SN soft interventions, identifiability up to

Table 1: Comparison of the results to existing work in multi-node interventional CRL. We note that all studies assume linear transformation.

| Work | Latent model | Int. type | Main assumption on interventions | Identifiability (ID) |
|------|-------------|-----------|----------------------------------|---------------------|
| [14] | Linear | Soft | $|\mathrm{pa}(i)|$ independent int. mechanisms | ID up to surrounding |
| [16] | General | do | strongly separated interventions | perfect ID |
| **Theorem 1** | General | Hard | lin. indep. interv. (Assumption 1) | perfect ID |
| **Theorem 2** | General | Soft | lin. indep. interv. (Assumption 1) | ID up to ancestors |

ancestors is shown for general latent models and linear transformations [7] and linear Gaussian latent models and general transformations [8]. Furthermore, under additional assumptions such as sufficiently nonlinear latent models, the latent DAG is shown to be perfectly identifiable [7, 10, 14]. In a related study, [15] focuses on learning the latent DAG (but without learning latent causal variables) using SN hard interventions without parametric assumptions on the model.

**Multi-node interventional CRL.** The studies on MN intervention settings are sparser than the SN intervention settings. Table 1 summarizes the results closely related to the scope of this paper along with the identifiability results established in this paper. In summary, the existing studies either provide *partial* identifiability or focus on non-stochastic do interventions. The study in [14] focuses on linear non-Gaussian latent models and linear transformations and uses soft interventions to establish identifiability up to surrounding variables by using multiple interventional mechanisms for each node. In a different study, [16] uses strongly separated multi-node do interventions and provides perfect identifiability results for general latent models and linear transformations. We also note the partially related study in [17] that applies soft interventions on a subset of nodes and aims to disentangle the non-intervened variables from the intervened ones. Distinct from all these studies, we address the open problem of perfect identifiability under UMN stochastic interventions.

**Other approaches to CRL.** We note that there exist other interesting settings that address CRL without interventions. Some examples include using multi-view data [18–21], leveraging temporal sequences [22, 23], building on nonlinear independent component analysis (ICA) principles to identify polynomial latent causal models [24], and imposing sparsity constraints to obtain partial disentanglement [25, 26], and grouping of observational variables [27]. We refer to [28] for a detailed literature review on various CRL problems.

## 2 CRL setting and preliminaries

**Notations.** Vectors are represented by lowercase bold letters, and element $i$ of vector $\mathbf{a}$ is denoted by $\mathbf{a}_i$. Matrices are represented by uppercase bold letters, and we denote row $i$ and column $j$ of matrix $\mathbf{A}$ by $\mathbf{A}_i$ and by $\mathbf{A}_{:,j}$, respectively, and $\mathbf{A}_{i,j}$ denotes the entry at row $i$ and column $j$. We use $\mathrm{null}(\{\mathbf{A}_1, \ldots, \mathbf{A}_r\})$ to denote the nullspace of the matrix consisting of the row vectors $\{\mathbf{A}_1, \ldots, \mathbf{A}_r\}$. For $n \in \mathbb{N}$, we define $[n] \triangleq \{1, \ldots, n\}$. The row permutation matrix associated with any permutation $\pi$ of $[n]$ is denoted by $\mathbf{P}_\pi$. We denote the indicator function by $\mathbb{1}$. We use $\mathrm{im}(f)$ to denote the image of a function $f$ and $\dim(\mathcal{V})$ to denote the dimension of a subspace $\mathcal{V}$. Random variables and their realizations are presented by upper and lower case letters, respectively.

### 2.1 Latent causal model

Consider a latent causal space consisting of $n$ causal random variables $Z \triangleq [Z_1, \ldots, Z_n]^\top$. An *unknown* linear transformation $\mathbf{G} \in \mathbb{R}^{d \times n}$ maps $Z$ to the observed random variables denoted by $X \triangleq [X_1, \ldots, X_d]^\top$ according to:

$$X = \mathbf{G} \cdot Z , \tag{1}$$

where $d \geq n$ and $\mathbf{G}$ is full rank. The probability density functions (pdfs) of $X$ and $Z$ are denoted by $p_X$ and $p_Z$, respectively. We assume that $p_Z$ has full support on $\mathbb{R}^n$. Subsequently, $p_X$ is supported on $\mathcal{X} \triangleq \mathrm{im}(\mathbf{G})$. The causal relationships among latent variables $Z$ are represented by a DAG $\mathcal{G}$ in which the $i$-th node corresponds to $Z_i$. Hence, $p_Z$ factorizes according to:

$$p_Z(z) = \prod_{i=1}^{n} p_i(z_i \mid z_{\mathrm{pa}(i)}) , \tag{2}$$

where $\mathrm{pa}(i)$ denotes the set of parents of node $i$ in $\mathcal{G}$. The conditional pdfs $\{p_i(z_i \mid z_{\mathrm{pa}(i)}) : i \in [n]\}$ are assumed to be continuously differentiable with respect to all $z$ variables. We use $\mathrm{ch}(i)$, $\mathrm{an}(i)$, and $\mathrm{de}(i)$ to denote the children, ancestors, and descendants of node $i$, respectively. We say that a permutation $(\pi_1, \ldots, \pi_n)$ of $[n]$ is a valid causal order if the membership $\pi_i \in \mathrm{an}(\pi_j)$ indicates that $i < j$. Without loss of generality, we assume that $(1, \ldots, n)$ is a valid causal order. We will specialize some of our results for the latent causal models with additive noise [1] specified by

$$Z_i = f_i(Z_{\mathrm{pa}(i)}) + N_i \,, \tag{3}$$

where functions $\{f_i : i \in [n]\}$ capture the causal dependence of node $i$ on its parents and the terms $\{N_i : i \in [n]\}$ represent the exogenous noise variables.

## 2.2   Unknown multi-node intervention models

In addition to the observational environment, we have $M$ UMN *interventional* environments denoted by $\{\mathcal{E}^m : m \in [M]\}$. We assume that the set of nodes intervened in each environment is *unknown*, and denote the set intervened in environment $\mathcal{E}^m$ by $I^m \subseteq [n]$. Accordingly, we define the *intervention signature matrix* $\mathbf{D}_{\mathrm{int}} \in \{0,1\}^{n \times M}$ to compactly represent the intervention targets under various environments as

$$[\mathbf{D}_{\mathrm{int}}]_{i,m} = \mathbb{1}\{i \in I^m\} \,, \quad \forall i \in [n] \,, \ \forall m \in [M] \,. \tag{4}$$

The $m$-th column of $\mathbf{D}_{\mathrm{int}}$ lists the indices of the nodes intervened in environment $\mathcal{E}^m$, which we refer to as the *intervention vector* of environment $\mathcal{E}^m$. Ensuring identifiability inevitably imposes restrictions on the structure of $\mathbf{D}_{\mathrm{int}}$. For instance, if the $i$-th row of $\mathbf{D}_{\mathrm{int}}$ is a zero vector, it means that node $i$ is not intervened in any environment, then the perfect identifiability is not possible [4] [2]. Therefore, to avoid such impossibility cases, we impose the mild condition that $\mathbf{D}_{\mathrm{int}}$ has sufficiently diverse columns, formalized next.

**Assumption 1.** *Intervention signature matrix* $\mathbf{D}_{\mathrm{int}}$ *defined in* (4) *is full row rank, i.e., it contains* $n$ *linearly independent intervention vectors.*

In this paper, we consider UMN stochastic interventions and address identifiability results under both hard interventions as well as soft interventions as the most general form of intervention.

**Soft interventions.** A soft intervention on node $i$ alters the *observational causal mechanism* $p_i(z_i \mid z_{\mathrm{pa}(i)})$ to an *interventional causal mechanism* $q_i(z_i \mid z_{\mathrm{pa}(i)})$. Such a change occurs in node $i$ in all the environments $\mathcal{E}^m$ that contain node $i$, i.e., $i \in I^m$. Subsequently, the pdf of the latent variables in environment $\mathcal{E}^m$, denoted by $p_Z^m$, factorizes according to:

$$p_Z^m(z) \triangleq \prod_{i \in I^m} q_i(z_i \mid z_{\mathrm{pa}(i)}) \prod_{i \notin I^m} p_i(z_i \mid z_{\mathrm{pa}(i)}) \,, \quad \forall m \in [M] \,. \tag{5}$$

**Hard interventions.** Under a *hard* intervention on node $i$, the functional dependence of node $i$ on its parents is removed, and the observational causal mechanism $p_i(z_i \mid z_{\mathrm{pa}(i)})$ is changed to an interventional causal mechanism $q_i(z_i)$, independent of parents of node $i$.

To distinguish the observational and interventional data, we denote the latent and observed random variables in environment $\mathcal{E}^m$ by $Z^m$ and $X^m$, respectively. We note that interventions do not affect the transformation $\mathbf{G}$. Hence, in $\mathcal{E}^m$ we have $X^m = \mathbf{G} \cdot Z^m$ for all $m \in [M]$.

**Score functions.** The score function of a pdf is defined as the gradient of its logarithm. We denote the score functions associated with the distributions of $Z^m$ and $X^m$ by

$$\boldsymbol{s}_Z^m(z) \triangleq \nabla \log p_Z^m(z) \,, \quad \text{and} \quad \boldsymbol{s}_X^m(x) \triangleq \nabla \log p_X^m(x) \,, \quad \forall m \in [M] \,. \tag{6}$$

Note that, using the factorization in (5), $\boldsymbol{s}_Z^m$ decomposes as

$$\boldsymbol{s}_Z^m(z) = \sum_{i \in I^m} \nabla \log q_i(z_i \mid z_{\mathrm{pa}(i)}) + \sum_{i \notin I^m} \nabla \log p_i(z_i \mid z_{\mathrm{pa}(i)}) \,. \tag{7}$$

We denote the difference in score functions between interventional and observational environments by

$$\Delta \boldsymbol{s}_Z^m(z) \triangleq \boldsymbol{s}_Z^m(z) - \boldsymbol{s}_Z(z) \quad \text{and} \quad \Delta \boldsymbol{s}_X^m(x) \triangleq \boldsymbol{s}_X^m(x) - \boldsymbol{s}_X(x) \,, \quad \forall m \in [M] \,. \tag{8}$$

---

[1]Perfect identifiability results in the closely related literature are given for additive noise models [7, 4, 8, 16].
 [2][4, Proposition 5] shows that if the non-intervened node $i$ has at least one parent, then perfect identifiability is not possible.

## 2.3 Identifiability criteria

In CRL, we use observed variables $X$ to recover the true latent variables $Z$ and the latent causal graph $\mathcal{G}$. We denote a generic estimator of $Z$ given $X$ by $\hat{Z}(X) : \mathbb{R}^d \to \mathbb{R}^n$. We also consider a generic estimate of $\mathcal{G}$ denoted by $\hat{\mathcal{G}}$. To assess the fidelity of the estimates $\hat{Z}(X)$ and $\hat{\mathcal{G}}$ with respect to the ground truth $Z$ and $\mathcal{G}$, we provide the following well-known identifiability measures.

**Definition 1** (Identifiability). *For CRL under linear transformations, we define:*

1. **Perfect identifiability:** $\hat{\mathcal{G}}$ *and* $\mathcal{G}$ *are isomorphic, and the estimator* $\hat{Z}(X)$ *satisfies that*

$$\hat{Z}(X) = \mathbf{P}_\pi \cdot \mathbf{C}_s \cdot Z , \qquad \forall Z \in \mathbb{R}^n , \tag{9}$$

   *where* $\mathbf{C}_s \in \mathbb{R}^{n \times n}$ *is a* constant *diagonal matrix with nonzero diagonal entries and* $\mathbf{P}_\pi$ *is a row permutation matrix.*

2. **Identifiability up to ancestors:** $\hat{\mathcal{G}}$ *and transitive closure of* $\mathcal{G}$*, denoted by* $\mathcal{G}_{tc}$*, are isomorphic, and the estimator* $\hat{Z}(X)$ *satisfies that*

$$\hat{Z}(X) = \mathbf{P}_\pi \cdot \mathbf{C}_{an} \cdot Z , \qquad \forall Z \in \mathbb{R}^n , \tag{10}$$

   *where* $\mathbf{C}_{an} \in \mathbb{R}^{n \times n}$ *is a* constant *matrix with nonzero diagonal entries that satisfies* $[\mathbf{C}_{an}]_{i,j} = 0$ *for all* $j \notin \{an(i) \cup \{i\}\}$*, and* $\mathbf{P}_\pi$ *is a row permutation matrix.*

In the algorithm we will design, estimating $\hat{Z}(X)$ and $\hat{\mathcal{G}}$ are facilitated by estimating the inverse of the transformation $\mathbf{G}$, that is Moore-Penrose inverse $\mathbf{G}^\dagger \triangleq [\mathbf{G}^\top \cdot \mathbf{G}]^{-1} \cdot \mathbf{G}^\top$, which we refer to as the *true encoder*. To formalize the process of estimating the true encoder, we define $\mathcal{H}$ as the set of candidate encoders specified by $\mathcal{H} \triangleq \{\mathbf{H} \in \mathbb{R}^{n \times d} : \mathsf{rank}(\mathbf{H}) = n \text{ and } \mathbf{H}^\top \cdot \mathbf{H} \cdot X = X , \forall X \in \mathcal{X}\}$. Corresponding to any pair of observation $X$ and valid encoder $\mathbf{H} \in \mathcal{H}$, we define $\hat{Z}(X; \mathbf{H})$ as an *auxiliary* estimate of $Z$ generated as $\hat{Z}(X; \mathbf{H}) \triangleq \mathbf{H} \cdot X = (\mathbf{H} \cdot \mathbf{G}) \cdot Z$.

## 3 Identifiability under UMN interventions

In this section, we present the main identifiability and achievability results for CRL with UMN interventions and interpret them in the context of the recent results in the literature. We start by specifying the regularity conditions on the statistical models, which are needed to ensure sufficient statistical diversity and establish identifiability results for hard and soft UMN interventions. The constructive proofs of the results are based on CRL algorithms, the details of which are presented in Section 4. Complete proofs are deferred to Appendix A.

We note that the UMN setting subsumes SN interventions. Similarly to all the existing identifiability results from SN interventions, it is necessary to have *sufficient* statistical diversity created by the intervention models.[3] These conditions can be generally presented in the form of regularity conditions on the probability distributions. Specifically, a commonly adopted regularity condition (or its variations) in the SN intervention setting is that for every possible pair $(i, j)$ where $i \in [n], j \in \mathrm{pa}(i)$, the following term cannot be a constant function in $z$,

$$\frac{\partial}{\partial z_j} \left( \log \frac{p_i(z_i \mid z_{\mathrm{pa}(i)})}{q_i(z_i \mid z_{\mathrm{pa}(i)})} \right) \left[ \frac{\partial}{\partial z_i} \log \frac{p_i(z_i \mid z_{\mathrm{pa}(i)})}{q_i(z_i \mid z_{\mathrm{pa}(i)})} \right]^{-1} . \tag{11}$$

We present a counterpart of these conditions for UMN interventions, which involves one additional term to account for the effect of intervening on multiple nodes simultaneously.

**Definition 2** (Intervention regularity). *We say that an interventions are* regular *if for every possible triplet* $(i, j, c)$ *where* $i \in [n], j \in \mathrm{pa}(i)$ *and* $c \in \mathbb{Q}$*, the following ratio cannot be a constant function in* $z$

$$\frac{\partial}{\partial z_j} \left( \log \frac{p_i(z_i \mid z_{\mathrm{pa}(i)})}{q_i(z_i \mid z_{\mathrm{pa}(i)})} + c \cdot \log \frac{p_j(z_j \mid z_{\mathrm{pa}(j)})}{q_j(z_j \mid z_{\mathrm{pa}(j)})} \right) \left[ \frac{\partial}{\partial z_i} \log \frac{p_i(z_i \mid z_{\mathrm{pa}(i)})}{q_i(z_i \mid z_{\mathrm{pa}(i)})} \right]^{-1} . \tag{12}$$

---

[3]Some examples include Assumption 1 in [6], generic SN interventions in [4], no pure shift interventions condition in [8], and the genericity condition in [9].

Essentially, $\frac{\partial}{\partial z_j} \log \frac{p_i(z_i|z_{\mathrm{pa}(i)})}{q_i(z_i|z_{\mathrm{pa}(i)})}$ captures the effect of intervening on node $i$ on the *score* associated with node $j$. In our method, we will use combinations of score differences of multi-node environments. This regularity condition ensures that the effect of a multi-node intervention is not the same on the scores associated with different nodes. Given these properties, we establish perfect identifiability for CRL with linear transformations using UMN stochastic hard interventions.

**Theorem 1** (Identifiability under UMN hard interventions). *Under Assumption 1 and a latent model with additive noise,*

1. *perfect latent recovery is possible using regular UMN hard interventions; and*

2. *if the latent causal model satisfies adjacency-faithfulness, then perfect latent DAG recovery is possible using regular UMN hard interventions.*

Theorem 1 is the first perfect identifiability result using UMN *stochastic* hard interventions. In contrast, [16] establishes perfect latent recovery using highly more stringent do-interventions. Furthermore, Theorem 1 establishes the first perfect latent DAG recovery result (under any type of multi-node interventions) for nonparametric latent models. We note that the capability of handling nonparametric latent models stems from leveraging the score functions. Similar properties are demonstrated by the prior work on score-based CRL for SN interventions [7]. It is noteworthy that we use a total of $n+1$ environments whereas the study in [16] requires $2\lceil \log_2 n \rceil$ do interventions of strongly separating sets. However, we show that identifiability is *impossible* using strongly separating sets of UMN stochastic hard interventions (see Appendix A.6).

Next, we consider UMN *soft* interventions. Since soft interventions retain the ancestral dependence of the intervened node, in general, the identifiability guarantees for soft interventions are weaker than those of hard interventions. Next, we establish that UMN soft interventions guarantee identifiability up to ancestors for the general causal latent models and linear transformations.

**Theorem 2** (Identifiability under UMN soft interventions). *Under Assumption 1, identifiability up to ancestors is possible using regular UMN soft interventions.*

Identifiability up to ancestors has recently shown to be possible using SN soft interventions on general latent models [7]. Theorem 2 establishes the same identifiability guarantees without the restrictive assumption of SN interventions. Furthermore, Theorem 2 is significantly different from existing results for UMN soft interventions. Specifically, the study in [14] focuses on linear non-Gaussian latent models and requires $|\mathrm{pa}(i)| + 1$ distinct mechanisms for each node $i$. In contrast, Theorem 2 does not make parametric assumptions on latent variables and works with sufficiently diverse interventions described by Assumption 1 instead of requiring multiple interventional mechanisms for the same node.

## 4 UMN interventional CRL algorithm

In this section, we design the **U**nknown **M**ulti-node **I**nterventional (UMNI)-CRL algorithm that achieves identifiability guarantees presented in Section 3. This algorithm falls in the category of score-based frameworks for CRL [5, 7] and incorporates novel components to this framework that facilitate UMN interventions with provable guarantees. Our score-based approach uses the structural properties of score functions and their variations across different interventional environments to find reliable estimates for the true encoder $\mathbf{G}^\dagger$. The critical step involved is a process that can aggregate the score differences under the available interventional environments, which have entirely *unknown* intervention targets, and reconstruct the score differences for any desired hypothetical set of intervention targets. In particular, we establish that such desired score differences can be computed by aggregating the score differences available under the given UMN interventions. The proposed UMNI-CRL algorithm consists of four stages for implementing CRL. The properties of these stages also serve as the steps of constructive proof for identifiability results. We present the key algorithmic stages and their properties in the remainder of this section and defer their proofs to Appendix A.

**Stage 1: Basis score differences.** In the first stage, we compute score differences for each interventional environment and construct the *basis score difference* functions that are linearly independent. The purpose of these functions is to subsequently use them and reconstruct the score differences under any arbitrary hypothetical interventional environment. To this end, we use the following relationship between score functions of $X$ and $Z$.

---

**Algorithm 1** Unknown **M**ulti-node **I**nterventional (UMNI)-CRL

---

1: **Input:** Samples of $X$ from environment $\mathcal{E}^0$ and interventional environments $\{\mathcal{E}^m : m \in [M]\}$
2: **Stage 1:** Choose basis score differences and construct $\Delta \mathbf{S}_X$ using (17)

---

3: **Stage 2: Identifiability up to a causal order**
4: **for** $t \in (1, \ldots, n)$ **do**
5:     **for** $\mathbf{w} \in \mathcal{W}$ **do**                                                                                  $\triangleright$ $\mathcal{W}$ is specified in (18)
6:         $\mathcal{V} \leftarrow \text{proj}_{\text{null}(\{\mathbf{H}_i^* : i \in [t-1]\})} \text{im}(\Delta \mathbf{S}_X \cdot \mathbf{w})$
7:         **if** $\dim(\mathcal{V}) = 1$ **then**
8:             pick $\mathbf{v} \in \text{im}(\Delta \mathbf{S}_X \cdot \mathbf{w}) \setminus \text{span}(\{\mathbf{H}_i^* : i \in [t-1]\})$
9:             $\mathbf{H}_t^* \leftarrow \mathbf{v}/\|\mathbf{v}\|_2$   and   $[\mathbf{W}]_{:,t} \leftarrow \mathbf{w}$
10:             **break**
11: Stage 2 outputs: $\mathbf{H}^*$ and $\mathbf{W}$

---

12: **Stage 3: Identifiability up ancestors**
13: Initialize $\hat{\mathcal{G}}$ with empty graph over nodes $[n]$
14: **for** $t \in (n-1, \ldots, 1)$ **do**
15:     **for** $j \in (t+1, \ldots, n)$ **do**
16:         **if** $j \in \hat{\text{ch}}(t)$ **then**
17:             **continue**
18:         `if_parent` $\leftarrow$ True
19:         $\mathcal{M}_{t,j} \leftarrow [j-1] \setminus \{\hat{\text{ch}}(t) \cup \{t\}\}$
20:         **for** $(\alpha, \beta) \in \{-(\|\mathbf{W}_{:,j}\|_1, \ldots, \|\mathbf{W}_{:,j}\|_1)\} \times [\|\mathbf{W}_{:,t}\|_1]$ **do**
21:             $\mathbf{w}^* \leftarrow \alpha[\mathbf{W}]_{:,t} + \beta[\mathbf{W}]_{:,j}$
22:             $\mathcal{V} \leftarrow \text{proj}_{\text{null}(\{\mathbf{H}_i^* : i \in \mathcal{M}_{t,j}\})} \text{im}(\Delta \mathbf{S}_X \cdot \mathbf{w}^*)$
23:             **if** $\dim(\mathcal{V}) = 1$ **then**
24:                 pick $\mathbf{v} \in \text{im}(\Delta \mathbf{S}_X \cdot \mathbf{w}^*) \setminus \text{span}(\{\mathbf{H}_i^* : i \in \mathcal{M}_{t,j}\})$
25:                 $\mathbf{H}_j^* \leftarrow \mathbf{v}/\|\mathbf{v}\|_2$   and   $[\mathbf{W}]_{:,j} \leftarrow \mathbf{w}^*$
26:                 Set `if_parent` $\leftarrow$ False and **break**
27:         **if** `if_parent` is True **then**
28:             Add $t \rightarrow j$ and $t \rightarrow u$ to $\hat{\mathcal{G}}$ for all $u \in \hat{\text{de}}(j)$     $\triangleright$ edges to identified descendants
29: Stage 3 outputs: $\hat{\mathcal{G}}, \mathbf{H}^*$ and $\mathbf{W}$

---

30: **if** the interventions are hard **then**
31:     **Stage 4: Unmixing for hard interventions**
32:     **for** $t \in (2, \ldots, n)$ **do**                                          $\triangleright$ refine rows of $\mathbf{H}^*$ sequentially
33:         $\hat{Z} \leftarrow \hat{Z}(X; \mathbf{H}^*)$
34:         $\mathbf{u}^{\text{obs}} \leftarrow -\text{Cov}(\hat{Z}_t, \hat{Z}_{\hat{\text{an}}(t)}) \cdot [\text{Cov}(\hat{Z}_{\hat{\text{an}}(t)})]^{-1}$
35:         **for** $m \in \{i : \mathbf{W}_{i,t} \neq 0\}$ **do**                              $\triangleright$ searching for a suitable environment
36:             $\hat{Z}^m \leftarrow \hat{Z}^m(X; \mathbf{H}^*)$
37:             $\mathbf{u}^m \leftarrow -\text{Cov}(\hat{Z}_t^m, \hat{Z}_{\hat{\text{an}}(t)}^m) \cdot [\text{Cov}(\hat{Z}_{\hat{\text{an}}(t)}^m)]^{-1}$
38:             **if** $(\mathbf{u}^m \cdot \hat{Z}_{\hat{\text{an}}(t)}^m + \hat{Z}_t^m) \perp\!\!\!\perp \hat{Z}_{\hat{\text{an}}(t)}^m$ and $\mathbf{u}^m \neq \mathbf{u}^{\text{obs}}$ **then**
39:                 $\mathbf{H}_t^* \leftarrow \mathbf{H}_t^* + \mathbf{u}^m \cdot \mathbf{H}_{\hat{\text{an}}(t)}^*$       $\triangleright$ removing the effect of the ancestors on $Z_t$
40:                 **break**
41:     $\hat{Z} \leftarrow \hat{Z}(X; \mathbf{H}^*)$                    $\triangleright$ use recovered $Z$ in obs. env. for graph recovery
42:     **for** $t \in (1, \ldots, n)$ **do**
43:         **for** $j \in \hat{\text{ch}}(t)$ **do**
44:             **if** $\hat{Z}_t \perp\!\!\!\perp \hat{Z}_j \mid \{\hat{Z}_i : i \in \hat{\text{pa}}(j) \setminus \{t\}\}$ **then**
45:                 Remove $t \rightarrow j$ from $\hat{\mathcal{G}}$     $\triangleright$ removing the edges from the nonparent ancestors

---

46: **Return** $\hat{\mathcal{G}}$ and $\hat{Z}$

---

**Lemma 1** ([7, Corollary 2]). *Latent and observational score functions are related via $s_X(x) = [\mathbf{G}^\dagger]^\top \cdot s_Z(z)$, where $x = \mathbf{G} \cdot z$.*

Using Lemma 1 for the scores and score differences defined in (7) and (8), respectively, we have

$$\Delta s_X^m(x) = [\mathbf{G}^\dagger]^\top \cdot \Delta s_Z^m(z) \overset{(7)}{=} [\mathbf{G}^\dagger]^\top \cdot \sum_{i \in I^m} \nabla \log \frac{p_i(z_i \mid z_{\mathrm{pa}(i)})}{q_i(z_i \mid z_{\mathrm{pa}(i)})} \,. \tag{13}$$

We compactly represent the summands in the right-hand side of (13) by defining the matrix-valued function $\mathbf{\Lambda} : \mathbb{R}^n \to \mathbb{R}^{n \times n}$ with the entries

$$[\mathbf{\Lambda}(z)]_{j,i} \triangleq \frac{\partial}{\partial z_j} \log \frac{p_i(z_i \mid z_{\mathrm{pa}(i)})}{q_i(z_i \mid z_{\mathrm{pa}(i)})} \,, \quad \forall i,j \in [n] \,, \tag{14}$$

based on which (13) can be restated as

$$\Delta s_X^m(x) = [\mathbf{G}^\dagger]^\top \cdot \mathbf{\Lambda}(z) \cdot [\mathbf{D}_{\mathrm{int}}]_{:,m} \,. \tag{15}$$

We note that $[\mathbf{\Lambda}(z)]_{i,j}$ is constantly zero for $i \notin \{\mathrm{pa}(j) \cup \{j\}\}$, and $j$-th column of $\mathbf{\Lambda}$ is a function of the variables in $\{z_k : k \in \mathrm{pa}(j) \cup \{j\}\}$ which implies that the columns of $\mathbf{\Lambda}$ are linearly independent. Throughout the rest of the paper, we omit the arguments of the functions $\Delta s_X^m$ and $\mathbf{\Lambda}$ when the dependence is clear from the context. Note that $\mathbf{D}_{\mathrm{int}}$ has $n$ linearly independent columns (Assumption 1). Denote the indices of the independent columns by $\{b_1, \ldots, b_n\}$ and define the *basis intervention matrix* $\mathbf{D} \in \mathbb{R}^{n \times n}$ using these columns as

$$[\mathbf{D}]_{i,m} \triangleq [\mathbf{D}_{\mathrm{int}}]_{i,b_m} = \mathbb{1}\{i \in I^{b_m}\} \,, \quad \forall i, m \in [n] \,. \tag{16}$$

Subsequently, it can be readily verified that the score difference functions $\{\Delta s_X^m : m \in \{b_1, \ldots, b_n\}\}$ are also linearly independent by leveraging (15) and linearly independent columns of $\mathbf{\Lambda}$. Hence, these score difference functions are sufficient to reconstruct the remaining unavailable score difference functions. As such, $\{\Delta s_X^m : m \in \{b_1, \ldots, b_n\}\}$ serve as *basis score difference* functions. We stack these basis score differences, where each is a $d$-dimensional vector, to construct the matrix-valued function $\Delta \mathbf{S}_X : \mathcal{X} \to \mathbb{R}^{d \times n}$, which is used as the basis score difference matrix in the subsequent stages.

$$\Delta \mathbf{S}_X \triangleq \left[ \Delta s_X^{b_1}, \ldots, \Delta s_X^{b_n} \right] = [\mathbf{G}^\dagger]^\top \cdot \mathbf{\Lambda} \cdot \mathbf{D} \,. \tag{17}$$

Finally, we note that $\Delta \mathbf{S}_X$ is directly estimated from samples of $X$ via learning $\{\Delta s_X^{b_m} : m \in [n]\}$. Since $\Lambda$ encodes the score differences in latent space, it cannot be estimated directly. Furthermore, $\mathbf{D}$ is unknown, and (17) is given to emphasize the relationship between observed and latent score differences.

**Stage 2: Identifiability up to an unknown causal order.** We design a process that aggregates score differences of the UMN interventions and obtains a *partial* identifiability guarantee (identifiability up to an unknown causal order) as an intermediate step toward more accurate identifiability. Specifically, we linearly aggregate the columns of $\Delta \mathbf{S}_X$ such that those aggregate scores facilitate identifiability up to an unknown causal order. Such mixing of the columns is facilitated by computing $\Delta \mathbf{S}_X \cdot \mathbf{W}$, where the mixing matrix $\mathbf{W} \in \mathbb{R}^{n \times n}$ should be learned. Given the decomposition of $\Delta \mathbf{S}_X$ in (17), if we learn $\mathbf{W}$ such that $\mathbf{D} \cdot \mathbf{W}$ is upper triangular up to a row permutation, then we can subsequently learn an intermediate estimate $\mathbf{H}^*$ using the image of $\Delta \mathbf{S}_X$. This ensures identifiability up to an unknown causal order since rows of $\mathbf{H}^*$ will be equal to combinations of rows of $\mathbf{G}^\dagger$ up to a causal order.

We design an iterative process to sequentially learn the columns of $\mathbf{W}$. Specifically, at each iteration of Stage 2, we learn an integer-valued vector $\mathbf{w}$ such that the projection of $\mathrm{im}(\Delta \mathbf{S}_X \cdot \mathbf{w})$ onto the nullspace of the partially recovered encoder estimate becomes a one-dimensional subspace. To see why this procedure works, note that the function $\Delta \mathbf{S}_X \cdot \mathbf{w}$ is essentially a combination of SN latent score differences via (17), and the SN score difference $\nabla \log p_i(z_i \mid z_{\mathrm{pa}(i)}) - \nabla \log q_i(z_i \mid z_{\mathrm{pa}(i)})$ is a one-dimensional subspace if and only if the intervened node $i$ has no parents. By taking the projection of the $\mathrm{im}(\Delta \mathbf{S}_X \cdot \mathbf{w})$ onto the nullspace of the partially learned encoder while searching for a desired $\mathbf{w}$, we ensure that the final encoder estimate $\mathbf{H}^*$ of this stage will be full-rank. Finally, we use $\kappa$ to denote maximum determinant of a matrix in $\{0, 1\}^{(n-1) \times (n-1)}$, and show that the set

$$\mathcal{W} \triangleq \{-\kappa, \ldots, +\kappa\}^n \tag{18}$$

is guaranteed to contain such $\mathbf{w}$ vectors. The following result summarizes the guarantees of this procedure.

**Lemma 2.** *Under Assumption 1 and intervention regularity, outputs of Stage 2 of Algorithm 1 satisfy:*

1. $\mathbf{D} \cdot \mathbf{W}$ *has nonzero diagonal entries and is upper triangular up to a row permutation.*

2. $[\mathbf{H}^*]_t \in \mathrm{span}(\{[\mathbf{G}^\dagger]_{\pi_j} : j \in [t]\})$ *for all* $t \in [n]$ *and* $\{[\mathbf{H}^*]_t : t \in [n]\}$ *are linearly independent.*

*Proof:* See Appendix A.1. □

**Stage 3: Identifiability up to ancestors.** Next, we refine the outcome of Stage 2 to ensure identifiability up to ancestors by updating the columns of $\mathbf{W}$ such that the entries of $\mathbf{D} \cdot \mathbf{W}$ can be nonzero only for the coordinates that correspond to ancestor-descendant node pairs. For this purpose, we design Stage 3 of UMNI-CRL that iteratively updates the columns of $\mathbf{W}$. The key idea is that the edges in the transitive closure graph $\mathcal{G}_{\mathrm{tc}}$ can be determined by investigating the subspaces' dimensions similarly to Stage 2. Leveraging this property, for all node pairs that do not constitute an edge in $\mathcal{G}_{\mathrm{tc}}$, we aggregate the corresponding columns of $\mathbf{W}$ such that the corresponding entry of $\mathbf{D} \cdot \mathbf{W}$ will be zero. In Theorem 3, we show that the outputs of this stage achieve identifiability up to ancestors, that is $\hat{\mathcal{G}}$ is isomorphic to $\mathcal{G}_{\mathrm{tc}}$ and $\hat{Z}_i$ is a linear function of $\{Z_{\pi_j} : j \in \mathrm{an}(i) \cup \{i\}\}$ for all $i \in [n]$.

**Theorem 3.** *Under Assumption 1 and regular UMN soft interventions, outputs of Stage 3 of Algorithm 1 have the following properties.*

1. *The estimate* $\hat{Z}(X; \mathbf{H}^*)$ *satisfies identifiability up to ancestors.*

2. $\hat{\mathcal{G}}$ *and* $\mathcal{G}_{\mathrm{tc}}$ *are related through a graph isomorphism.*

*Proof:* See Appendix A.2. □

**Stage 4: Perfect identifiability via hard interventions.** In the case of hard interventions, we apply an unmixing procedure to further refine our estimates and achieve perfect identifiability. This stage consists of two steps. The first step relies on the property that the intervened node becomes independent of its non-descendants and updates rows of the encoder estimate sequentially. In the second step, we leverage the knowledge of ancestral relationships and use a small number of conditional independence tests to refine the graph estimate from transitive closure $\mathcal{G}_{\mathrm{tc}}$ to the true latent DAG $\mathcal{G}$. The following theorem summarizes the guarantees achieved by Algorithm 1.

**Theorem 4.** *Under Assumption 1 and regular UMN hard interventions for an additive noise model, outputs of Stage 4 of Algorithm 1 have the following properties.*

1. *Estimate* $\hat{Z}(X; \mathbf{H}^*)$ *satisfies perfect latent variable recovery.*

2. *If* $p_Z$ *is adjacency-faithful to* $\mathcal{G}$, *then* $\hat{\mathcal{G}}$ *and* $\mathcal{G}$ *are related through a graph isomorphism.*

*Proof:* See Appendix A.3. □

Finally, we note that the computational cost of UMNI-CRL is dominated by the cardinality of the search space for aggregating the score differences, e.g., in the worst-case, Stage 2 has $\mathcal{O}((2\kappa)^n)$ complexity. Therefore, the structure of the UMN interventions determines the complexity via its determinant. We elaborate on the computational complexity of the algorithm and the range of $\kappa$ in Appendix A.8.

## 5 Simulations

We empirically assess the performance of the UMNI-CRL algorithm for recovering the latent DAG $\mathcal{G}$ and latent variables $Z$. Implementation details and additional results are provided in Appendix B[4].

**Data generation.** To generate $\mathcal{G}$, we use Erdős-Rényi model with density 0.5 and $n \in \{4, 5, 6, 7, 8\}$ nodes. For the causal models, we adopt linear structural equation models (SEMs) with Gaussian noise. The nonzero edge weights of the linear SEMs are sampled from $\mathrm{Unif}(\pm[0.5, 1.5])$, and the noise terms are zero-mean Gaussian variables with variances $\sigma_i^2$ sampled from $\mathrm{Unif}([0.5, 1.5])$. For a soft intervention on node $i$, the edge weight vector of node $i$ is reduced by a factor of $1/2$, and for a hard intervention, the edge weights are set to zero. The variance of the noise term is reduced to $\sigma_i^2/4$ in both intervention types. We consider target dimensions $d \in \{10, 50\}$, generate 100 latent graphs for each $(n, d)$ pair, and generate $n_{\mathrm{s}} = 10^5$ samples of $Z$ from each environment. Transformation

---

[4]The codebase for the experiments can be found at `https://github.com/acarturk-e/umni-crl`.

Table 2: UMNI-CRL for a linear causal model with UMN interventions (mean $\pm$ standard error)

| | | UMN Soft | | | UMN Hard | | |
| $n$ | $d$ | $\mathrm{SHD}(\mathcal{G}_{\mathrm{tc}}, \hat{\mathcal{G}})$ | MCC | $\ell_{\mathsf{soft}}$ | $\mathrm{SHD}(\mathcal{G}, \hat{\mathcal{G}})$ | MCC | $\ell_{\mathsf{hard}}$ |
|---|---|---|---|---|---|---|---|
| 4 | 10 | $0.91 \pm 0.12$ | $0.95 \pm 0.01$ | $0.08 \pm 0.01$ | $0.75 \pm 0.11$ | $0.98 \pm 0.02$ | $0.13 \pm 0.02$ |
| 5 | 10 | $1.67 \pm 0.20$ | $0.93 \pm 0.01$ | $0.09 \pm 0.01$ | $1.65 \pm 0.11$ | $0.97 \pm 0.02$ | $0.13 \pm 0.02$ |
| 6 | 10 | $3.19 \pm 0.26$ | $0.92 \pm 0.01$ | $0.12 \pm 0.01$ | $3.12 \pm 0.25$ | $0.96 \pm 0.02$ | $0.12 \pm 0.02$ |
| 7 | 10 | $5.44 \pm 0.34$ | $0.90 \pm 0.01$ | $0.15 \pm 0.01$ | $5.36 \pm 0.35$ | $0.93 \pm 0.03$ | $0.15 \pm 0.03$ |
| 8 | 10 | $7.63 \pm 0.41$ | $0.89 \pm 0.01$ | $0.16 \pm 0.01$ | $9.70 \pm 0.52$ | $0.87 \pm 0.03$ | $0.20 \pm 0.03$ |
| 4 | 50 | $0.77 \pm 0.12$ | $0.96 \pm 0.01$ | $0.06 \pm 0.01$ | $0.66 \pm 0.10$ | $0.98 \pm 0.01$ | $0.13 \pm 0.02$ |
| 5 | 50 | $1.93 \pm 0.20$ | $0.93 \pm 0.01$ | $0.10 \pm 0.01$ | $1.80 \pm 0.19$ | $0.98 \pm 0.02$ | $0.13 \pm 0.01$ |
| 6 | 50 | $3.39 \pm 0.27$ | $0.92 \pm 0.01$ | $0.13 \pm 0.01$ | $3.05 \pm 0.25$ | $0.95 \pm 0.03$ | $0.13 \pm 0.01$ |
| 7 | 50 | $4.62 \pm 0.30$ | $0.91 \pm 0.01$ | $0.13 \pm 0.01$ | $6.12 \pm 0.34$ | $0.91 \pm 0.02$ | $0.16 \pm 0.01$ |
| 8 | 50 | $8.26 \pm 0.49$ | $0.90 \pm 0.01$ | $0.14 \pm 0.01$ | $9.01 \pm 0.53$ | $0.88 \pm 0.03$ | $0.28 \pm 0.02$ |

$\mathbf{G} \in \mathbb{R}^{d \times n}$ is randomly sampled under full-rank constraint, and observed variables are generated as $X = \mathbf{G} \cdot Z$. Finally, for each graph realization, intervention matrix $\mathbf{D}$ is chosen randomly among column permutations of full-rank $\{0, 1\}^{n \times n}$ matrices, which satisfies Assumption 1.

**Score functions.** The algorithm uses score differences between environment pairs. Since we use a linear Gaussian model, $X$ is also multivariate Gaussian, and its score function can be estimated by $s_X(x) = -\hat{\Theta} \cdot x$ in which $\hat{\Theta}$ is the sample estimate of the precision matrix of $X$. We note that the design of UMNI-CRL is agnostic to the choice of the estimator and can adopt any reliable score estimator for nonparametric distributions [29, 30].

**Graph recovery.** To assess the graph recovery, we report the structural Hamming distance (SHD) between the true and estimated DAGs. Recall that UMN hard and soft interventions ensure different levels of identifiability guarantees. Hence, we report the SHD between (i) transitive closure $\mathcal{G}_{\mathrm{tc}}$ and $\hat{\mathcal{G}}$ for soft interventions, and (ii) true DAG $\mathcal{G}$ and $\hat{\mathcal{G}}$ for hard interventions. Table 2 shows that latent graph recovery performance remains consistent for both soft and hard interventions when observed variables dimension $d$ increases from 10 to 50, which conforms to our expectations due to theoretical results. Note that the expected number of edges is $n(n - 1)/4$ since we set the density of random graphs to $0.5$. Hence, the increasing SHD is also unsurprising when the latent dimension $n$ increases from 4 to 8, and the performance remains reasonable at $n = 8$. Finally, we note that $n = 8$ is the largest latent graph size considered among the closely related SN intervention studies [4, 8, 7, 9].

**Latent variable recovery.** The estimates are given by $\hat{Z}(X; \mathbf{H}^*) = (\mathbf{H}^* \cdot \mathbf{G}) \cdot Z$. Hence, we scrutinize the effective mixing matrix $(\mathbf{H}^* \cdot \mathbf{G})$ and report the ratio of its *incorrect mixing entries* to the number of zeros in constant matrices $\mathbf{C}_{\mathrm{s}}$ and $\mathbf{C}_{\mathrm{an}}$ according to Definition 1, denoted by

$$\ell_{\mathsf{hard}} \triangleq \frac{\sum_{j \neq i} \mathbb{1}([\mathbf{H}^* \cdot \mathbf{G}]_{i,j} \neq 0)}{n^2 - n}, \quad \text{and} \quad \ell_{\mathsf{soft}} \triangleq \frac{\sum_{j \notin \overline{\mathrm{an}}(i)} \mathbb{1}([\mathbf{H}^* \cdot \mathbf{G}]_{i,j} \neq 0)}{n^2 - \sum_i |\overline{\mathrm{an}}(i)|}. \quad (19)$$

We also report the mean correlation coefficient (MCC) [31], which measures linear correlations between the estimated and ground truth latent variables and is commonly used in related work. Table 2 shows that the UMNI-CRL algorithm achieves strong MCC performance (over $0.90$) in all cases. Furthermore, the ratio of incorrect mixing entries remains less than $0.20$ for both soft and hard interventions This demonstrates a strong performance of the UMNI-CRL algorithm at recovering latent variables for as many as $n = 8$ latent variables even for observed variables dimension $d = 50$.

## 6 Discussion

In this paper, we established novel identifiability results using unknown multi-node (UMN) interventions for CRL under linear transformations. Specifically, we designed the provably correct UMNI-CRL algorithm, leveraging the structural properties of score functions across different environments. To facilitate identifiability, we introduced a sufficient condition for the set of UMN interventions, abstracted as having $n$ sufficiently diverse interventional environments. Investigating the necessary conditions for UMN interventions to enable identifiability remains an open problem. The main limitation is the assumption of linear transformations. Given existing results for general transformations using two SN interventions per node [6, 9], a promising direction for future work is extending our results to general transformations using UMN interventions with multiple interventional mechanisms per node.

**Acknowledgements and disclosure of funding**

This work was supported by IBM through the IBM-Rensselaer Future of Computing Research Collaboration.

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

# Linear Causal Representation Learning from Unknown Multi-node Interventions
## Appendices

## Table of Contents

## A  Proofs

We start this section by introducing the additional notations used in the subsequent proofs.

**Additional notations.**  We use $\{\mathbf{e}_i : i \in [n]\}$ to denote $n$-dimensional standard unit basis vectors. For a set $\mathcal{S} \in [n]$, we use $\mathbf{A}_{\mathcal{S}}$ to denote the submatrix of $\mathbf{A}$ constructed by the rows $\{\mathbf{A}_i : i \in \mathcal{S}\}$. For a valid encoder $\mathbf{H} \in \mathcal{H}$, we denote the score functions associated with the pdfs of $\hat{Z}(X; \mathbf{H})$ under environments $\mathcal{E}^0$ and $\mathcal{E}^m$ by $\boldsymbol{s}_{\hat{Z}}(\cdot; \mathbf{H})$ and $\boldsymbol{s}_{\hat{Z}}^m(\cdot; \mathbf{H})$ for all $m \in [M]$. In addition to the graph notations introduced in Section 2.1, we define

$$\overline{\mathrm{pa}}(i) \triangleq \mathrm{pa}(i) \cup \{i\}, \quad \overline{\mathrm{ch}}(i) \triangleq \mathrm{ch}(i) \cup \{i\}, \quad \text{and} \quad \overline{\mathrm{an}}(i) \triangleq \mathrm{an}(i) \cup \{i\} . \tag{20}$$

Next, we provide an auxiliary result that will be used throughout the proofs of the main results.

**Lemma 3.**  *For every pair $(i, j)$ where $i \in [n]$ and $j \in \mathrm{pa}(i)$, the following ratio function cannot be a constant in $z$*

$$\frac{[\boldsymbol{\Lambda}(z)]_{j,j}}{[\boldsymbol{\Lambda}(z)]_{i,i}} = \left[ \frac{\partial}{\partial z_j} \log \frac{p_j(z_j \mid z_{\mathrm{pa}(j)})}{q_j(z_j \mid z_{\mathrm{pa}(j)})} \right] \cdot \left[ \frac{\partial}{\partial z_i} \log \frac{p_i(z_i \mid z_{\mathrm{pa}(i)})}{q_i(z_i \mid z_{\mathrm{pa}(i)})} \right]^{-1} . \tag{21}$$

*Furthermore, the columns of $\boldsymbol{\Lambda}$ are linearly independent vector-valued functions.*

*Proof:*  See Appendix A.5.  □

We also restate the interventional regularity in terms of the $\boldsymbol{\Lambda}$ function defined in (14), which will help clarity in the subsequent proofs.

**Definition 2 (Intervention regularity)** *We say that an intervention on node $i$ is* regular *if for every possible triplet $(i, j, c)$ where $i \in [n], j \in \mathrm{pa}(i)$ and $c \in \mathbb{Q}$, the following ratio cannot be a constant function in $z$*

$$\frac{\partial}{\partial z_j} \left( \log \frac{p_i(z_i \mid z_{\mathrm{pa}(i)})}{q_i(z_i \mid z_{\mathrm{pa}(i)})} + c \cdot \log \frac{p_j(z_j \mid z_{\mathrm{pa}(j)})}{q_j(z_j \mid z_{\mathrm{pa}(j)})} \right) \left[ \frac{\partial}{\partial z_i} \log \frac{p_i(z_i \mid z_{\mathrm{pa}(i)})}{q_i(z_i \mid z_{\mathrm{pa}(i)})} \right]^{-1} . \tag{22}$$

*By definition of $\Lambda$, this means that for $(i, j, c)$ where $i \in [n], j \in \mathrm{pa}(i)$ and $c \in \mathbb{Q}$, the following ratio cannot be a constant function in $z$*

$$\frac{\left[ \Lambda(z) \right]_{j,i} + c \cdot \left[ \Lambda(z) \right]_{j,j}}{\left[ \Lambda(z) \right]_{i,i}} . \tag{23}$$

## A.1 Proof of Lemma 2

First, we show that the search space in Stage 1 of Algorithm 1, i.e., the set $\mathcal{W}$, has a certain property that will be needed for the rest of the proof. Recall the definition in (18)

$$\mathcal{W} \triangleq \{-\kappa, \ldots, +\kappa\}^n , \tag{24}$$

in which $\kappa$ denotes the maximum possible determinant of a matrix in $\{0, 1\}^{(n-1) \times (n-1)}$. We will show that for all $i \in [n]$, there exists $\mathbf{w} \in \mathcal{W}$ such that $\mathbb{1}([\mathbf{D} \cdot \mathbf{w}] \neq 0) = \mathbf{e}_i$. Let $\mathrm{adj}(\mathbf{D})$ be the cofactor matrix of $\mathbf{D}$, which is the matrix formed by the entries

$$[\mathrm{adj}(\mathbf{D})]_{i,j} \triangleq \det([\mathbf{D}]_{-i,-j}) , \tag{25}$$

where $[\mathbf{D}]_{-i,-j}$ is the first minor of $\mathbf{D}$ with $i$-th row and $j$-th columns are removed. Then, the inverse of $\mathbf{D}$ is given by

$$\mathbf{D}^{-1} = \frac{1}{\det(\mathbf{D})} \cdot [\mathrm{adj}(\mathbf{D})]^\top . \tag{26}$$

By multiplying both sides from left by $\mathbf{D}$ and rearranging we obtain

$$\mathbf{D} \cdot [\mathrm{adj}(\mathbf{D})]^\top = \det(\mathbf{D}) \cdot \mathbf{I}_{n \times n} , \tag{27}$$

where all matrices are integer-valued and $\mathbf{I}_{n \times n}$ denotes the $n$-dimensional identity matrix. The identity in (27) implies that

$$\mathbf{D} \cdot \mathbf{w}^* = \det(\mathbf{D}) \cdot \mathbf{e}_i , \quad \text{where} \quad \mathbf{w}^* = [\mathrm{adj}(\mathbf{D})]_{:,i}^\top . \tag{28}$$

Since entries of $\mathrm{adj}(\mathbf{D})$ are upper bounded by $\kappa$, we have $\mathbf{w}^* \in \mathcal{W}$ and we conclude that

$$\forall i \; \exists \mathbf{w} \in \mathcal{W} \quad \text{such that} \quad \mathbb{1}(\mathbf{D} \cdot \mathbf{w} \neq 0) = \mathbf{e}_i . \tag{29}$$

Next, we prove the lemma statements by induction as follows.

**Base case.** At the base case $t = 1$, we will show that $\mathbb{1}([\mathbf{D} \cdot \mathbf{W}]_{:,1} \neq 0) = \mathbf{e}_i$ and $\mathbf{H}_1^* \in \mathrm{span}([\mathbf{G}^\dagger]_i)$ for some root node $i$. Consider a vector $\mathbf{w} \in \mathcal{W}$ and denote $\mathbf{c} = \mathbf{D} \cdot \mathbf{w}$. Then, using (17),

$$\Delta \mathbf{S}_X \cdot \mathbf{w} = [\mathbf{G}^\dagger]^\top \cdot \Lambda \cdot \mathbf{D} \cdot \mathbf{w} = [\mathbf{G}^\dagger]^\top \cdot \Lambda \cdot \mathbf{c} . \tag{30}$$

In the base case, we investigate the dimension of $\mathrm{im}(\Delta \mathbf{S}_X \cdot \mathbf{w})$. Then, using (30), we have

$$\dim \left( \mathrm{im}(\Delta \mathbf{S}_X \cdot \mathbf{w}) \right) = \dim \left( \mathrm{im}([\mathbf{G}^\dagger]^\top \cdot \Lambda \cdot \mathbf{c}) \right) = \dim \left( \mathrm{im}(\Lambda \cdot \mathbf{c}) \right) . \tag{31}$$

(29) implies that there exists $\mathbf{w}^* \in \mathcal{W}$ that makes $\mathbb{1}(\mathbf{c} \neq 0) = \mathbf{e}_i$. Subsequently, if $i$ is a root node, only nonzero entry of the column $\Lambda_{:,i}$ is $\Lambda_{i,i}$ and we have

$$\dim \left( \mathrm{im}(\Delta \mathbf{S}_X \cdot \mathbf{w}) \right) = \dim \left( \mathrm{im}([\mathbf{G}^\dagger]^\top \cdot \Lambda_{:,i}) \right) = \dim \left( \mathrm{im}([\mathbf{G}^\dagger]_i \cdot \Lambda_{i,i}) \right) = 1 . \tag{32}$$

Hence, by searching $\mathbf{w}$ within $\mathcal{W}$, Algorithm 1 is guaranteed to find a vector $\mathbf{w}$ such that $\dim(\mathrm{im}(\Lambda \cdot \mathbf{c})) = 1$. Next, we show that if $\dim(\mathrm{im}(\Lambda \cdot \mathbf{c})) = 1$, then $\mathbb{1}(\mathbf{c} \neq 0) = \mathbf{e}_i$ for a root node $i$. We prove this as follows. Consider the set $\mathcal{A} = \{i : \mathbf{c}_i \neq 0\}$ and let $i$ be the youngest node in $\mathcal{A}$, i.e., $i$ has no

descendants within $\mathcal{A}$. Using the fact that $\boldsymbol{\Lambda}_{\ell,k} = 0$ for all $\ell \notin \overline{\mathrm{pa}}(k)$ and $\mathbf{c}_k = 0$ for all $k \in \mathrm{de}(i)$, we have

$$\frac{\boldsymbol{\Lambda}_\ell \cdot \mathbf{c}}{\boldsymbol{\Lambda}_i \cdot \mathbf{c}} = \frac{\sum_{k \in \overline{\mathrm{ch}}(\ell)} \mathbf{c}_k \cdot \boldsymbol{\Lambda}_{\ell,k}}{\sum_{k \in \overline{\mathrm{ch}}(i)} \mathbf{c}_k \cdot \boldsymbol{\Lambda}_{i,k}} = \frac{\sum_{k \in \overline{\mathrm{ch}}(\ell)} \mathbf{c}_k \cdot \boldsymbol{\Lambda}_{\ell,k}}{\mathbf{c}_i \cdot \boldsymbol{\Lambda}_{i,i}} , \quad \forall \ell \in [n] . \tag{33}$$

If $\mathbf{c}$ has multiple nonzero entries, let $\mathbf{c}_\ell$ denote the second youngest node in $\mathcal{A}$. If $j \notin \mathrm{pa}(i)$, then for $\ell = j$, (33) reduces to

$$\frac{\boldsymbol{\Lambda}_j \cdot \mathbf{c}}{\boldsymbol{\Lambda}_i \cdot \mathbf{c}} = \frac{\mathbf{c}_j \cdot \boldsymbol{\Lambda}_{j,j}}{\mathbf{c}_i \cdot \boldsymbol{\Lambda}_{i,i}} , \tag{34}$$

which is not constant due to Lemma 3. If $j \in \mathrm{pa}(i)$, then (33) reduces to

$$\frac{\boldsymbol{\Lambda}_j \cdot \mathbf{c}}{\boldsymbol{\Lambda}_i \cdot \mathbf{c}} = \frac{\mathbf{c}_j \cdot \boldsymbol{\Lambda}_{j,j} + \mathbf{c}_i \cdot \boldsymbol{\Lambda}_{j,i}}{\mathbf{c}_i \cdot \boldsymbol{\Lambda}_{i,i}} , \tag{35}$$

which is not constant due to interventional regularity. Hence, in either case, there exist multiple vectors in $\mathrm{im}(\boldsymbol{\Lambda} \cdot \mathbf{c})$ with distinct directions. Therefore, $\dim(\mathrm{im}(\boldsymbol{\Lambda} \cdot \mathbf{c})) = 1$ implies that $\mathbf{c}$ has only one nonzero entry $\mathbf{c}_i \neq 0$. Finally, if $i$ is not a root node, for $j \in \mathrm{pa}(i)$, (33) reduces to

$$\frac{\boldsymbol{\Lambda}_j \cdot \mathbf{c}}{\boldsymbol{\Lambda}_i \cdot \mathbf{c}} = \frac{\boldsymbol{\Lambda}_{j,i}}{\boldsymbol{\Lambda}_{i,i}} , \tag{36}$$

which is not constant due to interventional regularity. Therefore, $\dim(\mathrm{im}(\boldsymbol{\Lambda} \cdot \mathbf{c})) = 1$ implies that $\mathbb{1}(c \neq 0) = \mathbb{1}([\mathbf{D} \cdot \mathbf{W}]_{:,1} \neq 0) = \mathbf{e}_i$ for a root node $i$. Next, using (30), we have

$$\Delta \mathbf{S}_X \cdot \mathbf{w} = \mathbf{c}_i \cdot \boldsymbol{\Lambda}_{i,i} \cdot [\mathbf{G}^\dagger]_i . \tag{37}$$

Hence, denoting the root node $i$ by $\pi_1$, setting $\mathbf{H}_1^* = \mathbf{v}$ for any $\mathbf{v} \in \mathrm{im}(\Delta \mathbf{S}_X \cdot \mathbf{w})$ ensures that $[\mathbf{D} \cdot \mathbf{W}]_{:,1} = \mathbf{e}_{\pi_1}$ and $\mathbf{H}_1^* \in \mathrm{span}([\mathbf{G}^\dagger]_{\pi_1})$ which completes the base step of the induction.

**Induction hypothesis.** For the induction step, assume that for all $t \in [k]$, we have linearly independent vectors $\mathbf{H}_t^* \in \mathrm{span}(\{[\mathbf{G}^\dagger]_{\pi_j} : j \in \overline{\mathrm{pa}}(t)\})$ and

$$[\mathbf{D} \cdot \mathbf{W}]_{\pi_j,t} = \begin{cases} 0 , & j > t \\ \neq 0 , & j = t \end{cases} , \quad \forall t \in [k], \ \forall j \geq t , \tag{38}$$

where $\{\pi_1, \ldots, \pi_k\}$ constitutes the first $k$ nodes of a causal order $\pi$. We will show that if

$$\dim(\mathcal{V}) = 1 \quad \text{where} \quad \mathcal{V} = \mathrm{proj}_{\mathsf{null}(\{\mathbf{H}_i^* : i \in [k]\})} \, \mathrm{im}(\Delta \mathbf{S}_X \cdot \mathbf{w}) , \tag{39}$$

then $\mathbf{w} \in \mathcal{W}$ satisfies

$$[\mathbf{D} \cdot \mathbf{w}]_{\pi_j} = \begin{cases} 0 , & j > k+1 \\ \neq 0 , & j = k+1 \end{cases} , \tag{40}$$

for the causal order $\pi$. First, define $\mathcal{M}_{k+1} \triangleq [n] \setminus \{\pi_1, \ldots, \pi_k\}$. Note that, by the induction premise,

$$\mathsf{null}(\{\mathbf{H}_i^* : i \in [k]\}) = \mathsf{null}(\{[\mathbf{G}^\dagger]_{\pi_i} : i \in [k]\}) . \tag{41}$$

Then, for any $\mathbf{w} \in \mathcal{W}$ and $\mathbf{c} = \mathbf{D} \cdot \mathbf{w}$, we have

$$\mathcal{V} = \mathrm{proj}_{\mathsf{null}(\{[\mathbf{G}^\dagger]_{\pi_i} : i \in [k]\})}([\mathbf{G}^\dagger]^\top \cdot \mathrm{im}(\boldsymbol{\Lambda} \cdot \mathbf{c})) . \tag{42}$$

Subsequently, we have

$$\dim(\mathcal{V}) = \dim \left( \mathrm{im} \left( [\mathbf{G}^\dagger]_{\mathcal{M}_{k+1}}^\top \cdot \boldsymbol{\Lambda}_{\mathcal{M}_{k+1}} \cdot \mathbf{c} \right) \right) = \dim \left( \mathrm{im}(\boldsymbol{\Lambda}_{\mathcal{M}_{k+1}} \cdot \mathbf{c}) \right) . \tag{43}$$

Note that, if node $i \in \mathcal{M}_{k+1}$ has no ancestors in $\mathcal{M}_{k+1}$, using the vector $\mathbf{w} \in \mathcal{W}$ which makes $\mathbb{1}(c \neq 0) = \mathbf{e}_i$ ensures that $\dim(\mathrm{im}(\boldsymbol{\Lambda}_{\mathcal{M}_{k+1}} \cdot)\mathbf{c}) = \dim(\mathrm{im}(\boldsymbol{\Lambda}_{i,i} \cdot \mathbf{e}_i)) = 1$. Hence, by searching $\mathbf{w}$ within $\mathcal{W}$, Algorithm 1 is guaranteed to find a vector $\mathbf{w}$ such that $\dim(\mathrm{im}(\boldsymbol{\Lambda}_{\mathcal{M}_{k+1}} \cdot \mathbf{c})) = 1$. Next, consider the set $\mathcal{A} = \{i \in \mathcal{M}_{k+1} : \mathbf{c}_i \neq 0\}$. Since $\pi_1, \ldots, \pi_k$ are first $k$-nodes of a causal order $\pi$, we have

$$\boldsymbol{\Lambda}_{i,\pi_j} = 0 , \quad \forall i \in \mathcal{M}_{k+1}, \ \forall j \in [k] . \tag{44}$$

Then, if $\dim(\mathcal{V}) = 1$, $\mathcal{A}$ is not empty since otherwise $(\mathbf{\Lambda}_{\mathcal{M}_{k+1}} \cdot \mathbf{c})$ is equal to zero vector and $\dim(\mathcal{V}) = 0$. Let $i$ be youngest node in $\mathcal{A}$, i.e, $i$ has no descendants within $\mathcal{A}$. Similarly to (33), we have

$$\frac{\mathbf{\Lambda}_\ell \cdot \mathbf{c}}{\mathbf{\Lambda}_i \cdot \mathbf{c}} = \frac{\sum_{k \in \overline{\mathrm{ch}}(\ell)} \mathbf{c}_k \cdot \mathbf{\Lambda}_{\ell,k}}{\mathbf{c}_i \cdot \mathbf{\Lambda}_{i,i}} , \quad \forall \ell \in \mathcal{M}_{k+1} . \tag{45}$$

If $\mathcal{A}$ has multiple elements, let $j$ be the second youngest node in $\mathcal{A}$. If $j \notin \mathrm{pa}(i)$, then for $\ell = j$, (45) reduces to

$$\frac{\mathbf{\Lambda}_j \cdot \mathbf{c}}{\mathbf{\Lambda}_i \cdot \mathbf{c}} = \frac{\mathbf{c}_j \cdot \mathbf{\Lambda}_{j,j}}{\mathbf{c}_i \cdot \mathbf{\Lambda}_{i,i}} , \tag{46}$$

which is not constant due to Lemma 3. If $j \in \mathrm{pa}(i)$, then for $\ell = j$, (45) reduces to

$$\frac{\mathbf{\Lambda}_j \cdot \mathbf{c}}{\mathbf{\Lambda}_i \cdot \mathbf{c}} = \frac{\mathbf{c}_j \cdot \mathbf{\Lambda}_{j,j} + \mathbf{c}_i \cdot \mathbf{\Lambda}_{j,i}}{\mathbf{c}_i \cdot \mathbf{\Lambda}_{i,i}} , \tag{47}$$

which is not constant due to interventional regularity. Hence, in either case, there exist vectors with different directions in $\mathrm{im}(\mathbf{\Lambda}_{\mathcal{M}_{k+1}} \cdot \mathbf{c})$. Therefore, $\dim(\mathrm{im}(\mathbf{\Lambda}_{\mathcal{M}_{k+1}} \cdot \mathbf{c})) = 1$ implies that $\mathcal{A}$ has only one element, i.e., there is only one node $i \in [n] \setminus \{\pi_1, \dots, \pi_k\}$ such that $\mathbf{c}_i \neq 0$, and also $\mathrm{pa}(i) \subseteq \{\pi_1, \dots, \pi_k\}$. Hence, denoting $\pi_{k+1} = i$, $[\mathbf{D} \cdot \mathbf{W}]_{\pi_j}$ is nonzero for $j = k+1$ and zero for all $j > k+1$. Subsequently, by setting $\mathbf{H}_{k+1}^* = v$ for any $\mathbf{v} \in \mathrm{im}(\Delta\mathbf{S}_X \cdot \mathbf{w}) \setminus \mathrm{span}(\{\mathbf{H}_i^* : i \in [k]\})$ we have

$$\mathbf{H}_{k+1}^* \in \mathrm{span}(\{[\mathbf{G}^\dagger]_{\pi_i} : i \in [k+1]\}) . \tag{48}$$

Finally, note that the contribution of $[\mathbf{G}^\dagger]_i$ in $\mathbf{H}_{k+1}^*$ is nonzero since $\mathbf{c}_i \neq 0$. Hence, $\mathbf{H}_{k+1}^*$ is linearly independent of $\{\mathbf{H}_1^*, \dots, \mathbf{H}_k^*\}$, which completes the proof of the induction hypothesis. Therefore, (38) holds for all $t \in [n]$, which implies that $\mathbf{P}_\pi \cdot \mathbf{D} \cdot \mathbf{W}$ is upper triangular and has nonzero diagonal entries, and concludes the proof of the lemma.

## A.2 Proof of Theorem 3

We will prove the theorem by proving the following equivalent statements: The output $\mathbf{W}$ of Stage 3 of Algorithm 1 satisfies

$$[\mathbf{D} \cdot \mathbf{W}]_{\pi_t, j} = \begin{cases} 0 , & \pi_t \notin \overline{\mathrm{an}}(\pi_j) \\ 1 , & \pi_t = \pi_j \end{cases} , \tag{49}$$

and

$$t \in \hat{\mathrm{pa}}(j) \iff \pi_t \in \mathrm{an}(\pi_j) . \tag{50}$$

for all $t, j \in [n]$. The second statement of the theorem, graph recovery up to a transitive closure, is equivalent to (50) by definition. Next, we show that (49) implies the first statement of the theorem, that is identifiability of latent variables up to mixing with ancestors. For this purpose, using (17), for any $\mathbf{w} \in \mathbb{R}^n$, we have

$$\Delta\mathbf{S}_X \cdot \mathbf{w} = [\mathbf{G}^\dagger]^\top \cdot \mathbf{\Lambda} \cdot \mathbf{D} \cdot \mathbf{w} . \tag{51}$$

Then, for $\mathbf{w} = [\mathbf{W}]_{:,j}$, using the fact that $\mathbf{\Lambda}_{\pi_t, \pi_k} = 0$ for all $\pi_t \notin \overline{\mathrm{pa}}(\pi_k)$, the coefficient of $[\mathbf{G}^\dagger]_{\pi_t}$ on the right-hand side becomes

$$\sum_{\pi_k \in [n]} \mathbf{\Lambda}_{\pi_t, \pi_k} \cdot [\mathbf{D} \cdot \mathbf{W}]_{\pi_k, j} = \sum_{\pi_k \in \overline{\mathrm{ch}}(\pi_t)} \mathbf{\Lambda}_{\pi_t, \pi_k} \cdot [\mathbf{D} \cdot \mathbf{W}]_{\pi_k, j} . \tag{52}$$

If $\pi_t \notin \overline{\mathrm{an}}(\pi_j)$, then $\pi_k \in \overline{\mathrm{ch}}(\pi_t)$ implies that $\pi_k \notin \overline{\mathrm{an}}(\pi_j)$. Then, using (49), the sum in (52), i.e., the coefficient of $[\mathbf{G}^\dagger]_{\pi_t}$ in function $(\Delta\mathbf{S}_X \cdot \mathbf{w})$, becomes zero. Furthermore, since $[\mathbf{D} \cdot \mathbf{W}]_{\pi_t, t} \neq 0$, the coefficient of $[\mathbf{G}^\dagger]_{\pi_t}$ in $(\Delta\mathbf{S}_X \cdot \mathbf{w})$ is nonzero. Therefore, (49) ensures, by choosing $\mathbf{H}_t^* \in \mathrm{im}(\Delta\mathbf{S}_X \cdot [\mathbf{W}]_{:,t})$ for all $t \in [n]$, we have

$$\mathbf{H}_t^* \in \mathrm{span}\left(\{[\mathbf{G}^\dagger]_{\pi_j} : j \in \overline{\mathrm{an}}(t)\}\right) \tag{53}$$

and $\{\mathbf{H}_1^*, \dots, \mathbf{H}_n^*\}$ are linearly independent. This implies that we have

$$\mathbf{H}^* = \mathbf{P}_\pi \cdot \mathbf{C}_{\mathrm{an}} \cdot \mathbf{G}^\dagger , \tag{54}$$

where $\mathbf{P}_\pi$ is the row permutation matrix of $\pi$, and $\mathbf{C}_{\mathrm{an}} \in \mathbb{R}^{n \times n}$ is a constant matrix with nonzero diagonal entries that satisfies $[\mathbf{C}_{\mathrm{an}}]_{i,j} = 0$ for all $j \notin \overline{\mathrm{an}}(i)$. Subsequently,

$$\hat{Z}(X; \mathbf{H}^*) = \mathbf{H}^* \cdot X = \mathbf{P}_\pi \cdot \mathbf{C}_{\mathrm{an}} \cdot \mathbf{G}^\dagger \cdot \mathbf{G} \cdot Z = \mathbf{P}_\pi \cdot \mathbf{C}_{\mathrm{an}} \cdot Z , \tag{55}$$

which is the definition of identifiability up to ancestors for latent variables, specified in Definition 1. In the rest of the proof, we will prove (49) by induction.

**Base case.** At the base case, we have $t = n - 1$ and $j = n$. We will show that, at the end of step $(t, j) = (n - 1, n)$, $[\mathbf{D} \cdot \mathbf{W}]_{\pi_{n-1}, n} \neq 0$ implies that $\pi_{n-1} \in \mathrm{pa}(\pi_n)$. By Lemma 2, we know that

$$[\mathbf{D} \cdot \mathbf{W}]_{\pi_{n-1}, n-1} \neq 0 \,, \quad [\mathbf{D} \cdot \mathbf{W}]_{\pi_n, n} \neq 0 \,, \quad \text{and} \quad [\mathbf{D} \cdot \mathbf{W}]_{\pi_n, n-1} = 0 \,. \tag{56}$$

Consider some integers $\alpha \in \{-n\kappa, \ldots, n\kappa\}$ and $\beta \in \{1, \ldots, n\kappa\}$. Let $\mathbf{w}^* = \alpha [\mathbf{W}]_{:, n-1} + \beta [\mathbf{W}]_{:, n}$ and $\mathbf{c}^* = \mathbf{D} \cdot \mathbf{w}^*$. Since $\beta \neq 0$, (56) implies that $\mathbf{c}_n^* \neq 0$. Using Lemma 2, we have

$$\mathsf{null}(\{\mathbf{H}_i^* \,:\, i \in [n-2]\}) = \mathsf{null}(\{[\mathbf{G}^\dagger]_{\pi_i} \,:\, i \in [n-2]\}) \,. \tag{57}$$

Then, we have

$$\mathcal{V} = \mathrm{proj}_{\mathsf{null}(\{\mathbf{H}_i^* : i \in [n-2]\})} \, \mathrm{im}(\Delta \mathbf{S}_X \cdot \mathbf{w}^*) \tag{58}$$

$$= \mathrm{proj}_{\mathsf{null}(\{[\mathbf{G}^\dagger]_{\pi_i} : i \in [n-2]\})} ([\mathbf{G}^\dagger]^\top \cdot \mathrm{im}(\mathbf{\Lambda} \cdot \mathbf{c}^*)) \,. \tag{59}$$

Subsequently,

$$\dim(\mathcal{V}) = \dim \left( \begin{bmatrix} [\mathbf{G}^\dagger]_{\pi_{n-1}} \\ [\mathbf{G}^\dagger]_{\pi_n} \end{bmatrix}^\top \cdot \mathrm{im} \left( \begin{bmatrix} \mathbf{\Lambda}_{\pi_{n-1}} \\ \mathbf{\Lambda}_{\pi_n} \end{bmatrix} \cdot \mathbf{c}^* \right) \right) \tag{60}$$

$$= \dim \left( \mathrm{im} \left( \begin{bmatrix} \mathbf{\Lambda}_{\pi_{n-1}} \\ \mathbf{\Lambda}_{\pi_n} \end{bmatrix} \cdot \mathbf{c}^* \right) \right) \tag{61}$$

$$= \dim \left( \mathrm{im} \left( \begin{bmatrix} \mathbf{c}_{\pi_{n-1}}^* \cdot \mathbf{\Lambda}_{\pi_{n-1}, \pi_{n-1}} + \mathbf{c}_{\pi_n}^* \cdot \mathbf{\Lambda}_{\pi_{n-1}, \pi_n} \\ \mathbf{c}_{\pi_n}^* \cdot \mathbf{\Lambda}_{\pi_n, \pi_n} \end{bmatrix} \right) \right) \,. \tag{62}$$

Note that in the last step, we have used the fact that $\pi$ is a causal order and $j \notin \overline{\mathrm{pa}}(i)$ implies that $\mathbf{\Lambda}_{i,j} = 0$. Then, if $\pi_{n-1} \in \mathrm{pa}(\pi_n)$, interventional regularity ensures that the ratio of the two entries in (62) is not a constant function which implies that $\dim(\mathcal{V}) = 2$. Furthermore, if $\pi_{n-1} \notin \mathrm{pa}(\pi_n)$ but $\mathbf{c}_{\pi_{n-1}}^* \neq 0$, then (62) reduces to

$$\dim(\mathcal{V}) = \dim \left( \mathrm{im} \left( \begin{bmatrix} \mathbf{c}_{\pi_{n-1}}^* \cdot \mathbf{\Lambda}_{\pi_{n-1}, \pi_{n-1}} \\ \mathbf{c}_{\pi_n}^* \cdot \mathbf{\Lambda}_{\pi_n, \pi_n} \end{bmatrix} \right) \right) = 2 \,, \tag{63}$$

due to Lemma 3. Therefore, if $\mathbf{w}^*$ satisfies

$$\dim(\mathrm{proj}_{\mathsf{null}(\{\mathbf{H}_i^* : i \in [n-2]\})} \, \mathrm{im}(\Delta \mathbf{S}_X \cdot \mathbf{w}^*)) = 1 \,, \tag{64}$$

by setting $[\mathbf{W}]_{:, n} = \mathbf{w}^*$, we guarantee that $[\mathbf{D} \cdot \mathbf{W}]_{\pi_{n-1}, n} = 0$ if $\pi_{n-1} \notin \mathrm{pa}(\pi_n)$. Note that, for this case, setting either $(\alpha, \beta) = (-[\mathbf{D} \cdot \mathbf{W}]_{n-1, n}, [\mathbf{D} \cdot \mathbf{W}]_{n-1, n-1})$ or $(\alpha, \beta) = ([\mathbf{D} \cdot \mathbf{W}]_{n-1, n}, -[\mathbf{D} \cdot \mathbf{W}]_{n-1, n-1})$ achieves

$$\mathbf{c}_{\pi_{n-1}}^* = [\mathbf{D} \cdot \mathbf{w}^*]_{\pi_{n-1}} = \alpha \cdot [\mathbf{D} \cdot \mathbf{W}]_{\pi_{n-1}, n-1} + \beta \cdot [\mathbf{D} \cdot \mathbf{W}]_{\pi_{n-1}, n} = 0 \,. \tag{65}$$

Note that the entries of $[\mathbf{D} \cdot \mathbf{W}]$ are bounded as

$$[\mathbf{D} \cdot \mathbf{W}]_{i,j} = \sum_{k \in [n]} \mathbf{D}_{i,k} \cdot \mathbf{W}_{k,j} \leq \sum_{k \in [n]} |\mathbf{W}_{k,j}| = \|\mathbf{W}_{:,j}\|_1 \,, \quad \forall i, j \in [n] \,. \tag{66}$$

Hence, Algorithm 1 is guaranteed to find $\mathbf{c}_{\pi_{n-1}}^* = 0$ by searching over

$$(\alpha, \beta) \in \{-\|\mathbf{W}_{:, n-1}\|_1, \ldots, \|\mathbf{W}_{:, n-1}\|_1\} \times \{1, \ldots, \|\mathbf{W}_{:, n}\|_1\} \,. \tag{67}$$

Therefore, if $\dim(\mathcal{V})$ is never found to be 1 for any $(\alpha, \beta)$, it means that $\pi_{n-1} \in \mathrm{pa}(\pi_n)$, and the algorithm adds the edge $(n - 1) \to n$, which concludes the proof of the base case.

**Induction hypothesis.** Next, assume that for all $t \in \{k + 1, \ldots, n\}$ and $j \in \{t + 1, \ldots, n\}$,

$$[\mathbf{D} \cdot \mathbf{W}]_{\pi_t, j} = \begin{cases} 0 & \pi_t \notin \overline{\mathrm{an}}(\pi_j) \\ 1 & \pi_t = \pi_j \end{cases} \quad \text{and} \quad \pi_t \in \mathrm{an}(\pi_j) \iff t \in \hat{\mathrm{pa}}(j) \,. \tag{68}$$

We will prove that (68) holds for $t = k$ and $j \in \{t + 1, \ldots, n\}$. We prove this by induction as well.

**Base case for the inner induction.** At the base case, we have $t = k$ and $j = k + 1$. Since $\pi$ is a causal order, we have

$$\pi_k \in \text{an}(\pi_{k+1}) \iff \pi_k \in \text{pa}(\pi_{k+1}) . \tag{69}$$

Also, $\hat{\text{ch}}(k)$ is empty at this iteration of the algorithm. Then, $\mathcal{M}_{k,k+1} = [k-1]$. Consider some integers $\alpha \in \{-n\kappa, \dots, n\kappa\}$ and $\beta \in \{1, \dots, n\kappa\}$. Let $\mathbf{w}^* = \alpha[\mathbf{W}]_{:,k} + \beta[\mathbf{W}]_{:,k+1}$ and $\mathbf{c}^* = \mathbf{D} \cdot \mathbf{w}^*$. Using Lemma 2, we have

$$\text{null}\big(\{\mathbf{H}_i^* : i \in [k-1]\}\big) = \text{null}\big(\{[\mathbf{G}^\dagger]_{\pi_i} : i \in [k-1]\}\big) . \tag{70}$$

Then, we have

$$\mathcal{V} = \text{proj}_{\text{null}(\{\mathbf{H}_i^* : i \in [k-1]\})} \, \text{im}(\Delta \mathbf{S}_X \cdot \mathbf{w}^*) \tag{71}$$

$$= \text{proj}_{\text{null}(\{[\mathbf{G}^\dagger]_{\pi_i} : i \in [k-1]\})} \big([\mathbf{G}^\dagger]^\top \cdot \text{im}(\Lambda \cdot \mathbf{c}^*)\big) . \tag{72}$$

Using (68), $\mathbf{c}_{\pi_i}^* = 0$ for $i > k + 1$. Subsequently,

$$\dim(\mathcal{V}) = \dim\left(\text{im}\left(\begin{bmatrix} [\mathbf{G}^\dagger]_{\pi_k} \\ [\mathbf{G}^\dagger]_{\pi_{k+1}} \end{bmatrix}^\top \cdot \begin{bmatrix} \Lambda_{\pi_k} \\ \Lambda_{\pi_{k+1}} \end{bmatrix} \cdot \mathbf{c}^*\right)\right) = \dim\left(\text{im}\left(\begin{bmatrix} \Lambda_{\pi_k} \\ \Lambda_{\pi_{k+1}} \end{bmatrix} \cdot \mathbf{c}^*\right)\right) . \tag{73}$$

Also, recall that $\Lambda_{\pi_k, \pi_j} = 0$ for all $j < k$. Then,

$$\dim(\mathcal{V}) = \dim\left(\text{im}\left(\begin{bmatrix} \Lambda_{\pi_k} \\ \Lambda_{\pi_{k+1}} \end{bmatrix} \cdot \mathbf{c}^*\right)\right) = \dim\left(\text{im}\left(\begin{bmatrix} \mathbf{c}_{\pi_k}^* \cdot \Lambda_{\pi_k, \pi_k} + \mathbf{c}_{\pi_{k+1}}^* \cdot \Lambda_{\pi_k, \pi_{k+1}} \\ \mathbf{c}_{\pi_{k+1}}^* \cdot \Lambda_{\pi_{k+1}, \pi_{k+1}} \end{bmatrix}\right)\right) . \tag{74}$$

Using (68) again, we have $\mathbf{c}_{\pi_{k+1}}^* \neq 0$. Then, if $\pi_k \in \text{pa}(\pi_{k+1})$, interventional regularity ensures that the ratio of the two entries in (74) is not a constant function which implies that $\dim(\mathcal{V}) = 2$. Furthermore, if $\pi_k \notin \text{pa}(\pi_{k+1})$ but $\mathbf{c}_{\pi_k}^* \neq 0$, based on Lemma 3, (74) reduces to

$$\dim(\mathcal{V}) = \dim\left(\text{im}\left(\begin{bmatrix} \mathbf{c}_{\pi_k}^* \cdot \Lambda_{\pi_k, \pi_k} \\ \mathbf{c}_{\pi_{k+1}}^* \cdot \Lambda_{\pi_{k+1}, \pi_{k+1}} \end{bmatrix}\right)\right) = 2 . \tag{75}$$

Therefore, if we have

$$\dim(\text{proj}_{\text{null}(\{\mathbf{H}_i^* : i \in [k-1]\})} \, \text{im}(\Delta \mathbf{S}_X \cdot \mathbf{w}^*)) = 1 , \tag{76}$$

by setting $[\mathbf{W}]_{:,k+1} = \mathbf{w}^*$, we guarantee that $[\mathbf{D} \cdot \mathbf{W}]_{\pi_k, k+1} = 0$ if $\pi_k \notin \text{pa}(\pi_{k+1})$. Note that, similarly to the base case of outer induction, Algorithm 1 is guaranteed to find $\mathbf{c}_{\pi_{n-1}}^* = 0$ by searching over $(\alpha, \beta) \in \{-\|\mathbf{W}_{:,k}\|_1, \dots, \|\mathbf{W}_{:,k}\|_1\} \times \{1, \dots, \|\mathbf{W}_{:,k+1}\|_1\}$. Therefore, if $\dim(\mathcal{V})$ is never found to be 1 for any $(\alpha, \beta)$, it means $\pi_k \in \text{pa}(\pi_{k+1})$, and the algorithm adds the edge $t \to j$, which concludes the proof of the base case for the inner induction.

**Induction hypothesis for the inner induction.** Next, assume that for all $j \in \{k + 1, \dots, u\}$,

$$[\mathbf{D} \cdot \mathbf{W}]_{\pi_k, j} = \begin{cases} 0 & \pi_k \notin \overline{\text{an}}(\pi_j) \\ 1 & \pi_k = \pi_j \end{cases} \quad \text{and} \quad \pi_k \in \text{an}(\pi_j) \iff k \in \hat{\text{pa}}(j) . \tag{77}$$

We will prove that (77) holds for $j = u+1$ as well. We work with $\mathcal{M}_{k,u+1} = [u-1] \setminus \{\hat{\text{ch}}(k) \cup \{k\}\}$. Consider some $\mathbf{w}^* = \alpha[\mathbf{W}]_{:,k} + \beta[\mathbf{W}]_{:,u+1}$ in which $\beta \neq 0$ and let $\mathbf{c}^* = \mathbf{D} \cdot \mathbf{w}^*$. Due to the assumption in (77), ancestors of any node in $\mathcal{M}_{k,u+1}$ are contained in $\mathcal{M}_{k,u+1}$. Then, using Lemma 2 again, we have

$$\text{null}(\{\mathbf{H}_i^* : i \in \mathcal{M}_{k,u+1}\}) = \text{null}(\{[\mathbf{G}^\dagger]_{\pi_i} : i \in \mathcal{M}_{k,u+1}\}) . \tag{78}$$

Then, we have

$$\mathcal{V} = \text{proj}_{\text{null}(\{\mathbf{H}_i^* : i \in \mathcal{M}_{k,u+1}\})} \, \text{im}(\Delta \mathbf{S}_X \cdot \mathbf{w}^*) \tag{79}$$

$$= \text{proj}_{\text{null}(\{[\mathbf{G}^\dagger]_{\pi_i} : i \in \mathcal{M}_{k,u+1}\})}([\mathbf{G}^\dagger]^\top \cdot \text{im}(\Lambda \cdot \mathbf{c}^*)) , \tag{80}$$

and

$$\dim(\mathcal{V}) = \dim\big(\text{im}([\Lambda]_{[n] \setminus \mathcal{M}_{k,u+1}} \cdot \mathbf{c}^*)\big) \geq \dim\left(\text{im}\left(\begin{bmatrix} \Lambda_{\pi_k} \\ \Lambda_{\pi_{u+1}} \end{bmatrix} \cdot \mathbf{c}^*\right)\right) . \tag{81}$$

We will investigate $\dim(\mathcal{V})$ through the ratio

$$\frac{\mathbf{\Lambda}_{\pi_k} \cdot \mathbf{c}^*}{\mathbf{\Lambda}_{\pi_{u+1}} \cdot \mathbf{c}^*} \, . \tag{82}$$

Note that, using (77) and the fact that $\pi$ is a causal order, we know that

$$\mathbf{c}_{\pi_i} = 0 \, , \quad \forall \, i \in \{u+2, \ldots, n\} \cup \{\{k+1, \ldots, u\} \setminus \hat{\mathrm{an}}(u+1)\} \, , \tag{83}$$

$$\text{and} \quad \mathbf{\Lambda}_{\pi_j, \pi_i} = 0 \, , \quad \forall \, \pi_j \notin \overline{\mathrm{pa}}(\pi_i) \, . \tag{84}$$

Then, using (83) and (84), we have

$$\mathbf{\Lambda}_{\pi_{u+1}} \cdot \mathbf{c}^* = \sum_{\pi_i \in \overline{\mathrm{ch}}(\pi_{u+1})} \mathbf{c}^*_{\pi_i} \cdot \mathbf{\Lambda}_{\pi_{u+1}, \pi_i} = \mathbf{c}^*_{\pi_{u+1}} \cdot \mathbf{\Lambda}_{\pi_{u+1}, \pi_{u+1}} \, , \tag{85}$$

$$\mathbf{\Lambda}_{\pi_k} \cdot \mathbf{c}^* = \mathbf{c}^*_{\pi_k} \cdot \mathbf{\Lambda}_{\pi_k, \pi_k} + \mathbf{c}^*_{\pi_{u+1}} \cdot \mathbf{\Lambda}_{\pi_k, \pi_{u+1}} + \sum_{i \, : \, \pi_i \in \mathrm{ch}(\pi_k) \text{ and } i \in \hat{\mathrm{an}}(u+1)} \mathbf{c}^*_{\pi_i} \cdot \mathbf{\Lambda}_{\pi_k, \pi_i} \, . \tag{86}$$

First, note that if there exists $\ell$ such that $\pi_k \in \mathrm{an}(\pi_\ell)$ and $\pi_\ell \in \mathrm{an}(\pi_{u+1})$, then by the assumptions in (68) and (77), we already have that $k \in \hat{\mathrm{pa}}(\ell)$ and $\ell \in \hat{\mathrm{pa}}(u+1)$ which implies that $k \in \hat{\mathrm{pa}}(u+1)$. Then, we only need to consider the case where there does not exist such $\ell$. In this case, the summation in (86) is zero and we have

$$\frac{\mathbf{\Lambda}_{\pi_k} \cdot \mathbf{c}^*}{\mathbf{\Lambda}_{\pi_{u+1}} \cdot \mathbf{c}^*} = \frac{\mathbf{c}^*_{\pi_k} \cdot \mathbf{\Lambda}_{\pi_k, \pi_k} + \mathbf{c}^*_{\pi_{u+1}} \cdot \mathbf{\Lambda}_{\pi_k, \pi_{u+1}}}{\mathbf{c}^*_{\pi_{u+1}} \cdot \mathbf{\Lambda}_{\pi_{u+1}, \pi_{u+1}}} \, . \tag{87}$$

Next, if $\pi_k \in \mathrm{pa}(\pi_{u+1})$, interventional regularity ensures that this ratio is not a constant function which implies that $\dim(\mathcal{V}) \geq 2$. Furthermore, if $\pi_k \notin \mathrm{an}(\pi_{u+1})$ but $\mathbf{c}^*_{\pi_k} \neq 0$, then (87) reduces to

$$\frac{\mathbf{\Lambda}_{\pi_k} \cdot \mathbf{c}^*}{\mathbf{\Lambda}_{\pi_{u+1}} \cdot \mathbf{c}^*} = \frac{\mathbf{c}^*_{\pi_k} \cdot \mathbf{\Lambda}_{\pi_k, \pi_k}}{\mathbf{c}^*_{\pi_{u+1}} \cdot \mathbf{\Lambda}_{\pi_{u+1}, \pi_{u+1}}} \, , \tag{88}$$

which is not constant due to Lemma 3 and subsequently $\dim(\mathcal{V}) \geq 2$. Therefore, $\dim(\mathcal{V}) = 1$ implies that $\pi_k \notin \mathrm{an}(\pi_{u+1})$ and $\mathbf{c}^*_{\pi_k} = [\mathbf{D} \cdot \mathbf{W}]_{\pi_k, u+1} = 0$. Finally, similarly to the previous cases, Algorithm 1 is guaranteed to find such $(\alpha, \beta)$ that makes $\mathbf{c}^*_{\pi_k} = 0$ by searching over $(\alpha, \beta) \in \{-\|\mathbf{W}_{:,k}\|_1, \ldots, \|\mathbf{W}_{:,k}\|_1\} \times \{1, \ldots, \|\mathbf{W}_{:,u+1}\|_1\}$, e.g., $(\alpha, \beta) = (-[\mathbf{D} \cdot \mathbf{W}]_{k, u+1}, [\mathbf{D} \cdot \mathbf{W}]_{k,k})$. Then, the proof of the inner induction step, and subsequently, the outer induction step is concluded, and (49) holds true. Consequently, the proof of the theorem is completed.

### A.3 Proof of Theorem 4

We start with a short synopsis of the proof. Stage 4 of Algorithm 1 consists of two steps. The first step resolves the mixing with ancestors in recovered latent variables and the second step refines the estimated graph to the edges from non-parent ancestors to children nodes. The proof of the first step uses similar ideas to that of [7, Lemma 10]. Specifically, we use zero-covariance as a surrogate for independence and search for *unmixing* vectors to eliminate the mixing with ancestors. Since we do not have SN interventions unlike the setting in [7], we use additional proof techniques to identify an environment in which a certain node is intervened. In the graph recovery stage, we leverage the knowledge of ancestral relationships and use a small number of conditional independence tests to remove the edges from non-parent ancestors to the children nodes.

#### A.3.1 Recovery of the latent variables

First, by Theorem 3, for the output $\mathbf{H}^*$ of Stage 3 of Algorithm 1 we have

$$[\mathbf{H}^* \cdot \mathbf{G}] = \mathbf{P}_\pi \cdot \mathbf{L} \, , \tag{89}$$

for some lower triangular matrix $\mathbf{L} \in \mathbb{R}^{n \times n}$ such that $\mathbf{L}$ has non-zero diagonal entries and $\mathbf{L}_{i,j} = 0$ for all $j \notin \overline{\mathrm{an}}(i)$, and $\pi$ is a causal order. We will show that the output of Stage 4 satisfies that

$$\mathbb{1}\left([\mathbf{H}^* \cdot \mathbf{G}]_t\right) = \mathbf{e}^\top_{\pi_t} \, , \quad \forall \, t \in [n] \, , \tag{90}$$

which will imply

$$\hat{Z}(X; \mathbf{H}^*) = \mathbf{H}^* \cdot X = \mathbf{H}^* \cdot \mathbf{G} \cdot Z = \mathbf{P}_\pi \cdot \mathbf{C}_\mathrm{s} \cdot Z \, , \tag{91}$$

where $\mathbf{P}_\pi$ is the row permutation matrix of $\pi$ and $\mathbf{C}_s$ is a constant diagonal matrix with nonzero diagonal entries. Note that this is the definition of perfect identifiability for latent variables, specified in Definition 1. We prove that Stage 4 output $\mathbf{H}^*$ satisfies (90) as follows.

We start by noting that since $\pi_1$ is a root node in $\mathcal{G}$ as shown in Lemma 2, (89) implies

$$[\mathbf{H}^* \cdot \mathbf{G}]_1 = \mathbf{L}_{\pi_1, \pi_1} \cdot \mathbf{e}_{\pi_1}^\top \quad \text{and} \quad \mathbb{1}\left([\mathbf{H}^* \cdot \mathbf{G}]_1\right) = \mathbf{e}_{\pi_1}^\top , \tag{92}$$

and (90) is satisfied for $t = 1$. Next, assume that (90) is satisfied for all $t \in \{1, \ldots, k-1\}$, i.e.,

$$\mathbb{1}\left([\mathbf{H}^* \cdot \mathbf{G}]_t\right) = \mathbf{e}_{\pi_t}^\top , \quad \forall t \in [k-1] . \tag{93}$$

Consider $k$-th iteration of the algorithm in which we update $\mathbf{H}_k^*$ where $\{\mathbf{H}_i^* : i \in [k-1]\}$ already satisfy (90). We will prove that (90) will be satisfied in four steps. Consider any environment $\mathcal{E}^m$ and use shorthand $\hat{Z}^m$ for $\hat{Z}^m(X; \mathbf{H}^*)$.

**Step 1:** $\mathrm{Cov}(\mathbf{u} \cdot \hat{Z}_{\hat{\mathrm{an}}(k)}^m + \hat{Z}_k^m, \hat{Z}_{\hat{\mathrm{an}}(k)}^m) = \mathbf{0}$ **has a unique solution for u.** For any $i \in [k-1]$, (93) implies

$$\hat{Z}_i^m = [\mathbf{H}^* \cdot \mathbf{G}] \cdot Z^m = \mathbf{c}_i \cdot Z_{\pi_i}^m , \tag{94}$$

for some nonzero constants $\{\mathbf{c}_1, \ldots, \mathbf{c}_{k-1}\}$. Specifically, by defining

$$\hat{\mathrm{an}}(k) = \{\gamma_1, \ldots, \gamma_r\} \tag{95}$$

in which the nodes are topologically ordered, then

$$\hat{Z}_{\hat{\mathrm{an}}(k)}^m = \left[\hat{Z}_{\gamma_1}^m, \ldots, \hat{Z}_{\gamma_r}^m\right] = \left[\mathbf{c}_{\gamma_1} \cdot Z_{\pi_{\gamma_1}}^m, \ldots, \mathbf{c}_{\gamma_r} \cdot Z_{\pi_{\gamma_r}}^m\right] . \tag{96}$$

Note that $\mathrm{Cov}(\hat{Z}_{\hat{\mathrm{an}}(k)}^m)$ is invertible since the causal relationships among the entries in $\hat{Z}_{\hat{\mathrm{an}}(k)}^m$ are not deterministic. Then, we have

$$\mathrm{Cov}(\mathbf{u} \cdot \hat{Z}_{\hat{\mathrm{an}}(k)}^m + \hat{Z}_k^m, \hat{Z}_{\hat{\mathrm{an}}(k)}^m) = 0 \quad \Longleftrightarrow \quad \mathbf{u} = -\mathrm{Cov}(\hat{Z}_k^m, \hat{Z}_{\hat{\mathrm{an}}(k)}^m) \cdot [\mathrm{Cov}(\hat{Z}_{\hat{\mathrm{an}}(k)}^m)]^{-1} . \tag{97}$$

For the next step, using the additive noise model specified in (3), under hard interventions we have

$$Z_{\pi_k}^m = f_{\pi_k}^m(Z_{\mathrm{pa}(\pi_k)}) + N_{\pi_k}^m , \quad \text{where} \quad (f_{\pi_k}^m, N_{\pi_k}^m) \triangleq \begin{cases} (f_{\pi_k}, N_{\pi_k}) , & \pi_k \notin I^m \\ (0, \bar{N}_{\pi_k}) , & \pi_k \in I^m \end{cases} , \tag{98}$$

where $\bar{N}_{\pi_k}$ denotes the exogenous noise term under intervention. Let us use $f$ and $N$ as shorthands for $f_{\pi_k}^m$ and $N_{\pi_k}^m$.

**Step 2:** $\mathrm{Cov}(\mathbf{u} \cdot \hat{Z}_{\hat{\mathrm{an}}(k)}^m + \hat{Z}_k^m, \hat{Z}_{\hat{\mathrm{an}}(k)}^m) = \mathbf{0}$ **and** $\mathbf{u} \cdot \hat{Z}_{\hat{\mathrm{an}}(k)}^m \perp\!\!\!\perp \hat{Z}_{\hat{\mathrm{an}}(k)}^m$ **implies that** $f_{\pi_k}^m$ **cannot be nonlinear.** Define random variable $U \triangleq (\mathbf{u} \cdot \hat{Z}_{\hat{\mathrm{an}}(k)}^m + \hat{Z}_k^m)$. Suppose that $\mathrm{Cov}(U, \hat{Z}_{\hat{\mathrm{an}}(k)}^m) = 0$ and $U \perp\!\!\!\perp \hat{Z}_{\hat{\mathrm{an}}(k)}^m$. Recall that using (89), $\hat{Z}_k^m$ is a linear combination of the variables $\{Z_{\pi_i}^m : i \in \mathrm{an}(k)\}$ and $Z_{\pi_k}^m$, i.e.,

$$\hat{Z}_k^m = \mathbf{c}' \cdot \hat{Z}_{\hat{\mathrm{an}}(k)}^m + \mathbf{c}_0' \cdot Z_{\pi_k}^m , \tag{99}$$

where $\mathbf{c}' \in \mathbb{R}^{|\hat{\mathrm{an}}(k)|}$ and $\mathbf{c}_0'$ is a non-zero scalar. Then, $U$ can be restated as

$$U = \mathbf{u} \cdot \hat{Z}_{\hat{\mathrm{an}}(k)}^m + \hat{Z}_k^m = (\mathbf{u} + \mathbf{c}') \cdot \hat{Z}_{\hat{\mathrm{an}}(k)}^m + \mathbf{c}_0' \cdot (f(Z_{\mathrm{pa}(k)}^m) + N) . \tag{100}$$

Note that $U \perp\!\!\!\perp \hat{Z}_{\hat{\mathrm{an}}(k)}^m$ and (96) together imply that $U$ is independent of any function of the variables in $\{Z_{\pi_i}^m : i \in \mathrm{an}(k)\}$. Hence,

$$(\mathbf{u} + \mathbf{c}') \cdot \hat{Z}_{\hat{\mathrm{an}}(k)}^m + \mathbf{c}_0' \cdot f(Z_{\mathrm{pa}(k)}) \perp\!\!\!\perp (\mathbf{u} + \mathbf{c}') \cdot \hat{Z}_{\hat{\mathrm{an}}(k)}^m + \mathbf{c}_0'(f(Z_{\mathrm{pa}(k)}) + N) , \tag{101}$$

where the right-hand size is $U$ and the left-hand side is a function of $\{Z_{\pi_i}^m : i \in \mathrm{an}(k)\}$. Also, since $N$ is the exogenous noise variable associated with $Z_{\pi_k}$, we have

$$(\mathbf{u} + \mathbf{c}') \cdot \hat{Z}_{\hat{\mathrm{an}}(k)}^m + \mathbf{c}_0' \cdot f(Z_{\mathrm{pa}(k)}) \perp\!\!\!\perp \mathbf{c}_0' \cdot N . \tag{102}$$

(101) and (102) imply that $(\mathbf{u} + \mathbf{c}') \cdot \hat{Z}_{\hat{\mathrm{an}}(k)}^m + \mathbf{c}_0' \cdot f(Z_{\mathrm{pa}(k)})$ is a constant function of $\hat{Z}_{\hat{\mathrm{an}}(k)}^m$. Since $(\mathbf{u} + \mathbf{c}') \cdot \hat{Z}_{\hat{\mathrm{an}}(k)}^m$ is a linear function (or constant zero) and $\mathbf{c}_0' \neq 0$, $f$ cannot be a nonlinear function which concludes the proof of this step. Next, we consider two possible cases, $f$ is zero, i.e., $\pi_k \in I^m$ case, and $f$ is a linear function.

**Step 3:** $\pi_k \in I^m$ **and** $f = 0$. In this case, we have $Z^m_{\pi_k} = N$. Note that for $\mathbf{u} = -\mathbf{c}'$, (100) becomes

$$U = \mathbf{c}'_0 \cdot N = \mathbf{c}'_0 \cdot Z^m_{\pi_k} \,, \tag{103}$$

which implies

$$(\mathbf{H}^*_k + \mathbf{u} \cdot \mathbf{H}^*_{\hat{\text{an}}(k)}) \cdot \mathbf{G} \cdot Z^m = \hat{Z}^m_k + \mathbf{u} \cdot \hat{Z}^m_{\hat{\text{an}}(k)} = U = \mathbf{c}'_0 \cdot Z^m_{\pi_k} \tag{104}$$

$$\text{and} \qquad \mathbb{1}\big((\mathbf{H}^*_k + \mathbf{u} \cdot \mathbf{H}^*_{\hat{\text{an}}(k)}) \cdot \mathbf{G}\big) = \mathbf{e}^\top_{\pi_k} \,. \tag{105}$$

Also note that, any $\mathbf{u}$ vector that satisfies $U \perp\!\!\!\perp \hat{Z}^m_{\hat{\text{an}}(k)}$ also satisfies $\text{Cov}(U, \hat{Z}^m_{\hat{\text{an}}(k)}) = \mathbf{0}$. In Step 1, we have shown that $\text{Cov}(U, \hat{Z}^m_{\hat{\text{an}}(k)}) = \mathbf{0}$ has a unique solution. Therefore, $U \perp\!\!\!\perp \hat{Z}^m_{\hat{\text{an}}(k)}$ also has a unique solution, which we have found in (97). Hence, if $\pi_k \in I^m$, then Algorithm 1 updates $\mathbf{H}^*_k$ correctly.

**Step 4:** $\pi_k \notin I^m$ **and** $f$ **is linear.** Only remaining case to check is $\pi_k \notin I^m$ and $f$ is a linear function. Let

$$f(Z_{\text{pa}(\pi_k)}) \triangleq \mathbf{c}'' \cdot \hat{Z}^m_{\hat{\text{an}}(k)} \,, \tag{106}$$

in which $\mathbf{c}'' \in \mathbb{R}^{|\hat{\text{an}}(k)|}$ has nonzero entries only at the coordinates corresponding to $\text{pa}(\pi_k)$. Then, for $(\mathbf{u} + \mathbf{c}') \cdot \hat{Z}^m_{\hat{\text{an}}(k)} + \mathbf{c}'_0 \cdot f(Z_{\text{pa}(k)})$ to be a constant function, we need to have $(\mathbf{u} + \mathbf{c}' + \mathbf{c}'_0 \cdot \mathbf{c}'') = \mathbf{0}$. Note that, for $\mathbf{c}'' \neq \mathbf{0}$ case, i.e., $f \neq 0$ and $\pi_k \notin I^m$, we have already found $\mathbf{u}^{\text{obs}} = -\mathbf{c}' - \mathbf{c}'_0 \cdot \mathbf{c}''$ using the observational environment. Since we are searching for $\mathbf{u} = -\mathbf{c}'$, which is required to achieve scaling consistency, we compare the solution $\mathbf{u}^m$ at environment $\mathcal{E}^m$ to $\mathbf{u}^{\text{obs}}$ and if they are distinct, we update $\mathbf{H}^*_k$.

To sum up, Stage 4 updates $\mathbf{H}^*_k$ correctly by identifying an environment $\mathcal{E}^m$ in which $Z_{\pi_k}$ is intervened and eliminating the effect of $\mathbf{H}^*_{\hat{\text{an}}(k)}$. Finally, we note that such an environment is guaranteed to exist among $\{\mathcal{E}^m : \mathbf{W}_{m,k} \neq 0\}$. To see this, recall that $[\mathbf{D} \cdot \mathbf{W}]_{\pi_k, k} \neq 0$ due to (49), proven in Theorem 3. This implies that there exists $m$ such that $\mathbf{D}_{\pi_k, m} = 1$ and $\mathbf{W}_{m,k} \neq 0$, and $\mathbf{D}_{\pi_k, m} = 1$ implies that $\pi_k \in I^m$. This completes the proof of (90), and subsequently the proof of the perfect latent recovery.

### A.3.2 Recovery of the latent graph

After the first step of Stage 4, we have

$$\hat{Z}_t = \mathbf{c}_t \cdot Z_{\pi_t} \,, \quad \forall t \in [n] \,, \tag{107}$$

where $\{\mathbf{c}_t : t \in [n]\}$ are nonzero constants. Then,

$$\hat{Z}_t \perp\!\!\!\perp \hat{Z}_j \mid \{\hat{Z}_i : i \in \mathcal{S}\} \quad \Longleftrightarrow \quad Z_{\pi_t} \perp\!\!\!\perp Z_{\pi_j} \mid \{Z_{\pi_i} : i \in \mathcal{S}\} \,. \tag{108}$$

Consider node $t \in [n]$ and node $j \in \hat{\text{ch}}(t)$. If $\pi_t \in \text{pa}(\pi_j)$, given the adjacency-faithfulness assumption, (108) implies that $\hat{Z}_t$ and $\hat{Z}_j$ cannot be made conditionally independent for any conditioning set. On the other hand, note that for any set $\mathcal{S}$ that contains all the nodes in $\text{pa}(\pi_j)$ and does not contain a node in $\text{de}(\pi_j)$ satisfies

$$Z_{\pi_t} \perp\!\!\!\perp Z_{\pi_j} \mid \{Z_{\pi_i} : i \in \mathcal{S}\} \,, \tag{109}$$

and subsequently,

$$\hat{Z}_t \perp\!\!\!\perp \hat{Z}_j \mid \{\hat{Z}_i : i \in \mathcal{S}\} \,. \tag{110}$$

Finally, if $\pi_t \notin \text{pa}(\pi_j)$, then $\hat{\text{pa}}(j) \setminus \{t\}$ contains all the nodes in $\text{pa}(\pi_j)$ and does not contain a node in $\text{de}(\pi_j)$. Hence, the second stage of Step 4 of Algorithm 1 successfully eliminates all spurious edges between $t$ and $j \in \hat{\text{ch}}(t)$.

### A.4 Proofs of Theorem 1 and Theorem 2

Under Assumption 1 and interventional regularity, Lemma 2 and Theorem 3 show that using UMN soft interventions, outputs of Algorithm 1 satisfy identifiability up to ancestors. Hence, identifiability up to ancestors is possible using UMN soft interventions.

Furthermore, note that Lemma 2 and Theorem 3 are valid for both soft and hard interventions. Then, Theorem 4 shows that using UMN hard interventions, Algorithm 1 outputs satisfy perfect identifiability. Hence, perfect identifiability is possible using UMN hard interventions.

## A.5 Proof of Lemma 3

We start by showing that $[\Lambda(z)]_{i,i}$ cannot be a constant function in $z$. This is shown in the proof of [5, Lemma 7]. For our paper to be self-contained, we repeat the proof steps in [5] as follows. For $i \in [n]$ define

$$h(z_i, z_{\mathrm{pa}(i)}) \triangleq \frac{p_i(z_i \mid z_{\mathrm{pa}(i)})}{q_i(z_i \mid z_{\mathrm{pa}(i)})} . \tag{111}$$

We prove by contradiction that $h(z_i, z_{\mathrm{pa}(i)})$ varies with $z_i$. Assume the contrary, i.e., let $h(z_i, z_{\mathrm{pa}(i)}) = h(z_{\mathrm{pa}(i)})$. By rearranging (111) we have

$$p_i(z_i \mid z_{\mathrm{pa}(i)}) = h(z_{\mathrm{pa}(i)}) \cdot q_i(z_i \mid z_{\mathrm{pa}(i)}) . \tag{112}$$

Fix a realization of $z_{\mathrm{pa}(i)} = z^*_{\mathrm{pa}(i)}$, and integrate both sides of (112) with respect to $z_i$. Since both $p_i$ and $q_i$ are pdfs, we have

$$1 = \int_{\mathbb{R}} p_i(z_i \mid z^*_{\mathrm{pa}(i)}) \mathrm{d}z_i = \int_{\mathbb{R}} h(z^*_{\mathrm{pa}(i)}) \cdot q_i(z_i \mid z^*_{\mathrm{pa}(i)}) \mathrm{d}z_i \tag{113}$$

$$= h(z^*_{\mathrm{pa}(i)}) \int_{\mathbb{R}} q_i(z_i \mid z^*_{\mathrm{pa}(i)}) \mathrm{d}z_i \tag{114}$$

$$= h(z^*_{\mathrm{pa}(i)}) . \tag{115}$$

This identity implies that $p_i(z_i \mid z^*_{\mathrm{pa}(i)}) = q_i(z_i \mid z^*_{\mathrm{pa}(i)})$ for any arbitrary realization $z^*_{\mathrm{pa}(i)}$, which contradicts with the premise that observational and interventional causal mechanisms are distinct. Consequently,

$$[\Lambda(z)]_{i,i} = \frac{\partial}{\partial z_i} \log \frac{p_i(z_i \mid z_{\mathrm{pa}(i)})}{q_i(z_i \mid z_{\mathrm{pa}(i)})} \tag{116}$$

is not a constant in $z$. Note that for a fixed realization of $z_j = z^*_j$, $z_{\mathrm{pa}(i)} = z^*_{\mathrm{pa}(i)}$, and $z_{\mathrm{pa}(j)} = z^*_{\mathrm{pa}(j)}$, $[\boldsymbol{\Lambda}(z)]_{j,j}$ becomes constant whereas $[\boldsymbol{\Lambda}(z)]_{i,i}$ varies with $z_i$. Hence, their ratio is not a constant in $z$. Finally, note that $\boldsymbol{\Lambda}(z)$ is an upper-triangular matrix since $(1, \ldots, n)$ is a valid causal order and for all $i \in [n]$,

$$\nabla \log \frac{p_i(z_i \mid z_{\mathrm{pa}(i)})}{q_i(z_i \mid z_{\mathrm{pa}(i)})} \tag{117}$$

is a function of only $\{z_k : k \in \mathrm{pa}(i) \cup \{i\}\}$. Together with the fact that diagonal entries of $\boldsymbol{\Lambda}(z)$ are not constantly zero, the columns (and rows) of $\boldsymbol{\Lambda}$ are linearly independent vector-valued functions.

## A.6 Insufficiency of strongly separating sets

**Background.** Recently, [16] has shown the identifiability of latent representations under a linear transformation using do interventions. Specifically, they have shown that a *strongly separating set* of multi-node interventions are sufficient for identifiability. It is well-known that strongly separating sets can be constructed using $2\lceil \log_2 n \rceil$ elements. Hence, identifiability can be achieved using $2\lceil \log n \rceil$ do interventions. In this section, we show that a similar result is *impossible* when using stochastic hard interventions.

**Lemma 4** (Impossibility). *A strongly separated set of $2\lceil \log_2 n \rceil$ stochastic hard interventions are not guaranteed to be sufficient for perfect identifiability. In fact, they are not even sufficient for identifiability up to ancestors.*

*Proof:* We prove the claim for $n = 2$ nodes. The smallest strongly separating set for two nodes is $\{\{1\}, \{2\}\}$. We will consider two distinct models of latent variables and latent graphs that are not distinguishable using interventional data of $I^1 = \{1\}$ and $I^2 = \{2\}$ (without observational data). The key idea is that after the linear transformation in the first model is fixed, we can design the linear transformation in the second model such that observed variables in both models will have the same distributions. We construct a pair of indistinguishable models as follows.

**First model.** Let $\mathcal{G}$ consists of the edge $1 \to 2$. Consider a linear Gaussian latent model with the edge weight of $Z_1$ on $Z_2$ is set to 1, and consider identity mapping, i.e.,

$$\mathcal{G} : 1 \to 2 \,, \qquad \mathbf{G} = \begin{bmatrix} 1 & 0 \\ 0 & 1 \end{bmatrix} \,, \quad Z^1 = \begin{bmatrix} N_1^* \\ N_1^* + N_2 \end{bmatrix} \,, \quad Z^2 = \begin{bmatrix} N_1 \\ N_2^* \end{bmatrix} \,, \tag{118}$$

$$N_1 \sim \mathcal{N}(0, V_1) \,, \quad N_1^* \sim \mathcal{N}(0, V_1^*) \,, \quad N_2 \sim \mathcal{N}(0, V_2) \,, \quad N_2^* \sim \mathcal{N}(0, V_2^*) \,. \tag{119}$$

where $V_1, V_1^*, V_2, V_2^*$ are nonzero variances of the exogenous noise terms such that $\bar{V}_1 \neq \bar{V}_1^*$ to ensure that interventional and observational mechanisms of node 1 are distinct. Then, the observed variables $X$ in two environments are given by

$$X^1 = \mathbf{G} \cdot Z^1 \sim \mathcal{N}\left(0, \begin{bmatrix} V_1^* & V_1^* \\ V_1^* & V_1^* + V_2 \end{bmatrix}\right) \,, \tag{120}$$

$$\text{and} \quad X^2 = \mathbf{G} \cdot Z^2 \sim \mathcal{N}\left(0, \begin{bmatrix} V_1 & 0 \\ 0 & V_2^* \end{bmatrix}\right) \,. \tag{121}$$

**Second model.** Let $\bar{\mathcal{G}}$ be the empty graph. Consider a linear Gaussian latent model under a non-identity mapping parameterized by

$$\bar{\mathcal{G}} : \text{empty} \,, \qquad \bar{\mathbf{G}} = \begin{bmatrix} a & b \\ c & d \end{bmatrix} \,, \quad \bar{Z}^1 = \begin{bmatrix} \bar{N}_1^* \\ \bar{N}_2 \end{bmatrix} \,, \quad Z^2 = \begin{bmatrix} \bar{N}_1 \\ \bar{N}_2^* \end{bmatrix} \,, \tag{122}$$

$$\bar{N}_1 \sim \mathcal{N}(0, \bar{V}_1) \,, \quad \bar{N}_1^* \sim \mathcal{N}(0, \bar{V}_1^*) \,, \quad \bar{N}_2 \sim \mathcal{N}(0, \bar{V}_2) \,, \quad \bar{N}_2^* \sim \mathcal{N}(0, \bar{V}_2^*) \,. \tag{123}$$

where $\bar{V}_1, \bar{V}_1^*, \bar{V}_2, \bar{V}_2^*$ are nonzero variances of the exogenous noise terms such that $\bar{V}_1 \neq \bar{V}_1^*$ and $\bar{V}_2 \neq \bar{V}_2^*$ to ensure that interventional mechanisms are distinct from the observational ones. Then, the observed variables $\bar{X}$ in two environments are given by

$$\bar{X}^1 = \bar{\mathbf{G}} \cdot \bar{Z}^1 \sim \mathcal{N}\left(0, \begin{bmatrix} a^2 \bar{V}_1^* + b^2 \bar{V}_2 & ac\bar{V}_1^* + bd\bar{V}_2 \\ ac\bar{V}_1^* + bd\bar{V}_2 & c^2 \bar{V}_1^* + d^2 \bar{V}_2 \end{bmatrix}\right) \,, \tag{124}$$

$$\text{and} \quad \bar{X}^2 = \bar{\mathbf{G}} \cdot \bar{Z}^2 \sim \mathcal{N}\left(0, \begin{bmatrix} a^2 \bar{V}_1 + b^2 \bar{V}_2^* & ac\bar{V}_1 + bd\bar{V}_2^* \\ ac\bar{V}_1 + bd\bar{V}_2^* & c^2 \bar{V}_1 + d^2 \bar{V}_2^* \end{bmatrix}\right) \,. \tag{125}$$

**Non-identifiability.** We will show a nontrivial construction that ensures $X^1 = \bar{X}^1$ and $X^2 = \bar{X}^2$, which implies the non-identifiability from intervention set $\{\{1\}, \{2\}\}$. First, using (120), (121), (124), and (125), we write all requirements for $X^1 = \bar{X}^1$ and $X^2 = \bar{X}^2$ to hold:

$$V_1^* = a^2 \bar{V}_1^* + b^2 \bar{V}_2 \,, \tag{126}$$

$$V_1^* = ac\bar{V}_1^* + bd\bar{V}_2 \,, \tag{127}$$

$$V_2 = (c^2 - ac)\bar{V}_1^* + (d^2 - bd)\bar{V}_2 \,, \tag{128}$$

$$V_1 = a^2 \bar{V}_1 + b^2 \bar{V}_2^* \,, \tag{129}$$

$$V_2^* = c^2 \bar{V}_1 + d^2 \bar{V}_2^* \,, \tag{130}$$

$$0 = ac\bar{V}_1 + bd\bar{V}_2^* \,. \tag{131}$$

Now, let $\bar{V}_1, \bar{V}_1^*, \bar{V}_2, \bar{V}_2^*$ take any values such that $\bar{V}_1 \bar{V}_2 \neq \bar{V}_1^* \bar{V}_2^*$. We want to show that there exist $\{a, b, c, d\}$ and $\{V_1, V_1^*, V_2, V_2^*\}$ values that satisfy all the six equations. Note that, the values of $V_1, V_2,$ and $V_2^*$ are not constrained by multiple equations or additional conditions. Hence, for any given $\{a, b, c, d\}$ and $\{\bar{V}_1, \bar{V}_1^*, \bar{V}_2, \bar{V}_2^*\}$, we can readily set $V_1, V_2,$ and $V_2^*$ to satisfy (129), (128), and (130), respectively. Therefore, we only need to ensure that we can choose $\{a, b, c, d\}$ such that the identities in (126),(127), and (131) are satisfied. Next, after substituting $d = 1$, and rearranging, we only need to choose $\{a, b, c\}$ that satisfy

$$ac\bar{V}_1 + b\bar{V}_2^* = 0 \,, \tag{132}$$

$$(a^2 - ac)\bar{V}_1^* + (b^2 - b)\bar{V}_2 = 0 \,. \tag{133}$$

Substituting $ac = -b\dfrac{\bar{V}_2^*}{\bar{V}_1}$ into (133), we require

$$b^2 \cdot \bar{V}_2 - b(\bar{V}_2 - \bar{V}_2^* \frac{\bar{V}_1^*}{\bar{V}_1}) + a^2 \bar{V}_1^* = 0 \,. \tag{134}$$

(134) has a real solution for $b$ if and only if

$$a^2 \leq \frac{\bar{V}_2}{4\bar{V}_1^*} \left(1 - \frac{\bar{V}_1^* \bar{V}_2^*}{\bar{V}_1 \bar{V}_2}\right)^2 . \tag{135}$$

This implies that, if $\bar{V}_1 \bar{V}_2 \neq \bar{V}_1^* \bar{V}_2^*$, we can choose $a$ and $b$ that satisfies (134), and $c$ is determined by (132). This means that, for any given $\{\bar{V}_1, \bar{V}_1^*, \bar{V}_2, \bar{V}_2^*\}$ such that $\bar{V}_1 \bar{V}_2 \neq \bar{V}_1^* \bar{V}_2^*$, there exists $\{a, b, c, d\}$ and $\{V_1, V_1^*, V_2, V_2^*\}$ that ensures $X^1 = \bar{X}^1$ and $X^2 = \bar{X}^2$. Hence, interventions on the strongly separating set $\{\{1\}, \{2\}\}$ are not sufficient to distinguish the first and second models, which completes the proof of non-identifiability. $\qquad\square$

### A.7  Analysis of interventional regularity

**Lemma 5.** *Consider additive noise models under hard interventions. Interventional regularity is satisfied if the post-intervention score function of $N_i$, denoted by $\bar{r}$, is analytic and one of the following is true.*

*1. $\dfrac{\partial f_i(z_{\mathrm{pa}(i)})}{\partial z_j}$ is not constant and there do not exist constants $\alpha_1 \neq 1$, $\alpha_2, \alpha_3 \in \mathbb{R}$ such that*

$$\bar{r}(y) = \alpha_1 \cdot \bar{r}(y + \alpha_2) + \alpha_3 , \quad \forall y \in \mathbb{R} . \tag{136}$$

*2. $\dfrac{\partial f_i(z_{\mathrm{pa}(i)})}{\partial z_j}$ is constant and noise term $N_i$ remains unaltered after the intervention.*

*Proof:* The additive noise model for nodes $i$ and $j$ are given by

$$Z_i = f_i(Z_{\mathrm{pa}(i)}) + N_i , \quad \text{and} \quad Z_j = f_j(Z_{\mathrm{pa}(j)}) + N_j . \tag{137}$$

When nodes $i$ and $j$ are intervened, respectively, $Z_i$ and $Z_j$ are generated according to

$$Z_i = \bar{N}_i \quad \text{and} \quad Z_j = \bar{N}_j , \tag{138}$$

in which $\bar{N}_i$ and $\bar{N}_j$ correspond to exogenous noise terms for nodes $i$ and $j$ under intervention. Then, denoting the pdfs of $N_i, \bar{N}_i, N_j, \bar{N}_j$ by $h_i, \bar{h}_i, h_j, \bar{h}_j$, respectively, we have

$$p_i(z_i \mid z_{\mathrm{pa}(i)}) = h_i(z_i - f_i(z_{\mathrm{pa}(i)})) , \tag{139}$$
$$q_i(z_i) = \bar{h}_i(z_i) , \tag{140}$$
$$p_j(z_j \mid z_{\mathrm{pa}(j)}) = h_j(z_j - f_j(z_{\mathrm{pa}(j)})) , \tag{141}$$
$$q_j(z_j) = \bar{h}_j(z_j) . \tag{142}$$

Then, by denoting the score functions associated with $h_i, \bar{h}_i, h_j, \bar{h}_j$ by $r_i, \bar{r}_i, r_j, \bar{r}_j$, respectively, we have

$$\frac{\partial}{\partial z_i} \log \frac{p_i(z_i \mid z_{\mathrm{pa}(i)})}{q_i(z_i)} = r_i(n_i) - \bar{r}_i(n_i + f_i(z_{\mathrm{pa}(i)})) , \tag{143}$$

$$\frac{\partial}{\partial z_j} \log \frac{p_i(z_i \mid z_{\mathrm{pa}(i)})}{q_i(z_i)} = -r_i(n_i) \cdot \frac{\partial f_i(z_{\mathrm{pa}(i)})}{\partial z_j} , \tag{144}$$

$$\frac{\partial}{\partial z_j} \log \frac{p_j(z_j \mid z_{\mathrm{pa}(j)})}{q_j(z_j)} = r_j(n_j) - \bar{r}_j(n_j + f_j(z_{\mathrm{pa}(j)})) . \tag{145}$$

Next, assume the contrary and let the ratio in (12) be a constant $\alpha \in \mathbb{R}$. Then, substituting (143), (144), and (145) into (12) and rearranging the terms, we have

$$\left(\alpha + \frac{\partial f_i(z_{\mathrm{pa}(i)})}{\partial z_j}\right) \cdot r_i(n_i) - \alpha \cdot \bar{r}_i(n_i + f_i(z_{\mathrm{pa}(i)})) = c \cdot \left(r_j(n_j) - \bar{r}_j(n_j + f_j(z_{\mathrm{pa}(j)}))\right) . \tag{146}$$

**Case 1:** $\frac{\partial f_i(z_{\mathrm{pa}(i)})}{\partial z_j}$ **is not constant.** Consider two distinct realizations of $Z_{\mathrm{pa}(i) \cup \mathrm{pa}(j)}$ such that $\frac{\partial}{\partial z_j} f_i(z_{\mathrm{pa}(i)})$ takes values of $\beta_1$ and $\beta_2$ where $\beta_1 \neq \beta_2$. Then, (146) implies

$$(\alpha + \beta_1) \cdot r_i(n_i) - \alpha \cdot \bar{r}_i(n_i + \gamma_1) = c \cdot u_1 , \tag{147}$$

$$(\alpha + \beta_2) \cdot r_i(n_i) - \alpha \cdot \bar{r}_i(n_i + \gamma_2) = c \cdot u_2 , \tag{148}$$

for some constants $\gamma_1, \gamma_2, u_1, u_2$ for all $n_i \in \mathbb{R}$, and $\beta_1 \neq \beta_2$. By rearranging the terms, we get rid of $r_i(n_i)$ terms and obtain

$$\alpha \cdot ((\alpha + \beta_2) \cdot \bar{r}_i(n_i + \gamma_1) - (\alpha + \beta_1) \cdot \bar{r}_i(n_i + \gamma_2)) = c \cdot ((\alpha + \beta_1) \cdot u_2 - (\alpha + \beta_2) \cdot u_1) . \tag{149}$$

Since $\beta_1 \neq \beta_2$, (149) implies that

$$\bar{r}_i(y) = \alpha_1 \cdot \bar{r}_i(y + \alpha_2) + \alpha_3 , \tag{150}$$

where $\alpha_1, \alpha_2, \alpha_3$ are constants and $\alpha_1 \neq 1$. Therefore, if there does not exist such constants, then the ratio in (12) cannot be a constant.

**Case 2:** $\frac{\partial f_i(z_{\mathrm{pa}(i)})}{\partial z_j} = \beta$ **for some nonzero constant** $\beta$. In this case, (146) becomes

$$(\alpha + \beta) \cdot r_i(n_i) - \alpha \cdot \bar{r}_i(n_i + f_i(z_{\mathrm{pa}(i)})) = c \cdot \left( r_j(n_j) - \bar{r}_j(n_j + f_j(z_{\mathrm{pa}(j)})) \right) . \tag{151}$$

Note that the right-hand side of (151) does not contain $n_i$. Also note that since $f_i(z_{\mathrm{pa}(i)})$ is continuous, there exists an open interval $\Theta \subseteq \mathbb{R}$ such that $f_i(z_{\mathrm{pa}(i)})$ can take all values $\theta \in \Theta$. Then, by taking the derivative of both sides with respect to $n_i$ and varying $f_i(z_{\mathrm{pa}(i)})$ in $\Theta$, we find that $\bar{r}_i'(n_i + \theta)$ is constant for all $\theta \in \Theta$. Since $\bar{r}_i$ is analytic, this means that $\bar{r}_i(y) = \alpha_1 \cdot y + \alpha_2$ for some constants $\alpha_1, \alpha_2$ for all $y \in \mathbb{R}$. Then, since the noise term is invariant under intervention, we have $r_i = \bar{r}_i$. Substituting this into (151), we obtain

$$(\alpha + \beta) \cdot (\alpha n_i + \alpha_2) - \alpha \alpha_1 \cdot (n_i + f_i(z_{\mathrm{pa}(i)})) = c \cdot \left( r_j(n_j) - \bar{r}_j(n_j + f_j(z_{\mathrm{pa}(j)})) \right) . \tag{152}$$

Since $\beta \neq 0$, the left-hand side is a function of $n_i$ whereas the right-hand side is not, which is invalid. Hence, the ratio in (12) cannot be constant in this case. $\qquad \square$

### A.8 Computational complexity of UMNI-CRL algorithm

UMNI-CRL (Algorithm 1) consists of four stages that we elaborate on as follows.

**Stage 1:** The score difference estimation is only performed once before the subsequent main algorithm steps. Since our results and the algorithm do not rely on a specific score difference estimation technique, studying the computational complexity of this step is out of scope.

**Stage 2:** In this step, we check the dimension of $\mathcal{V}$, a projection of a subspace (see line 6 of Algorithm 1) at most $n \times (2\kappa + 1)^n$ times. Here, $\kappa$ denotes the maximum possible determinant of a matrix in $\{0, 1\}^{(n-1) \times (n-1)}$. In the proof of Lemma 2 in Appendix A.1, we discuss why this choice facilitates the identifiability guarantees.

**Stage 3:** In this step, we check the dimension of $\mathcal{V}$ (see line 23 of Algorithm 1) at most $\|\mathbf{W}_{:,j}\|_1 \times \|\mathbf{W}_{:,t}\|_1$ times for every $(t, j)$ in Stage 3. Note that $\mathbf{W}$ is updated within the steps of Stage 3. Therefore, the exact computational complexity would be a function of the graph structure and the outputs of Stage 2.

**Stage 4:** Unmixing procedure for hard interventions essentially operates as a post-processing step that does not pose additional computational challenges. For instance, the total number of conditional independence tests in Stage 4 is $\mathcal{O}(n^2)$.

**Bounds on** $\kappa$**.** In Section 4, we defined $\kappa$ as the maximum determinant of a binary matrix $\{0, 1\}^{(n-1) \times (n-1)}$ and noted that the computational complexity of UMNI-CRL algorithm depends on $n$ and $\kappa$ as seen in Stage 2 above. In general, using the well-known bound for the determinant of $\{0, 1\}$ matrices [32], we have

$$\kappa \leq \lfloor 2(n/4)^n \rfloor . \left| \left[ \mathrm{adj}(\mathbf{D}) \right]_{i,j} \right| \leq \frac{n^{n/2}}{2^{n-1}} = 2 \left( \frac{n}{4} \right)^n . \tag{153}$$

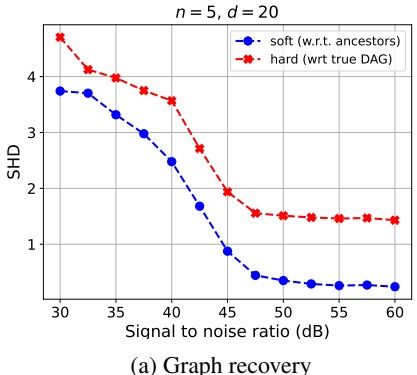

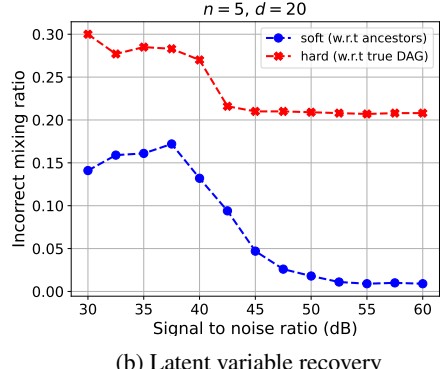

(a) Graph recovery

(b) Latent variable recovery

Figure 1: Sensitivity analysis of UMNI-CRL algorithm for quadratic latent causal models. The results are for $n = 5$ latent nodes and $d = 20$ observed variables, $10^4$ samples, and for average of 100 runs. **(a):** $\mathrm{SHD}(\mathcal{G}_{\mathrm{tc}}, \hat{\mathcal{G}})$ versus SNR (soft) and $\mathrm{SHD}(\mathcal{G}, \hat{\mathcal{G}})$ versus SNR (hard). **(b):** Incorrect mixing ratio $\ell_{\mathrm{soft}}$ versus SNR (soft) and $\ell_{\mathrm{hard}}$ versus SNR (hard).

However, for specific cases, we have much smaller upper bounds for $\kappa$. For instance, for the choices of $n$ being $2, 3, 4, 5, 6, 7$, $\kappa$ is known to be upper bounded by $1, 1, 2, 3, 5, 9$, respectively [32]. Furthermore, [33] shows that the determinant of a matrix in $\{0, 1\}^{n \times n}$ with $n + k$ nonzero entries is bounded by $2^{k/3}$. Hence, if $\mathbf{D}$ has $n + k$ nonzero entries, then

$$\kappa \leq \lfloor 2^{k/3} \rfloor . \tag{154}$$

This implies that $\kappa$ can be quite small for sparse UMN interventions. For instance, if we take the exhaustive set of SN interventions and only add two additional intervened nodes in total, then we have $\kappa = 1$.

**Empirical tricks.** Finally, we note that various empirical tricks can be used to reduce the algorithm's computational complexity. For instance, after every update to $\mathbf{W}$, we can divide the columns of $\mathbf{W}$ by the greatest common divisor of its entries. Furthermore, even though $\kappa$ can grow quickly as $n$ becomes larger, in our experiments with up to $n = 8$ nodes, we observe that setting $\kappa = 2$ usually suffices for a good performance. Hence, we set $\kappa = 2$ in the simulations in Section 5.

## B    Additional simulations

**Details of evaluation metrics.** For the graph recovery via hard interventions, we use conditional independence tests in Stage 4 of Algorithm 1. Since we adopt a linear Gaussian SEM latent model, we use a partial correlation test and set the significance level to $\alpha = 0.05$. For the latent variable recovery error metrics $\ell_{\mathrm{hard}}$ and $\ell_{\mathrm{soft}}$ defined in (19), we first pass the entries of $\mathbf{H}^* \cdot \mathbf{G}$ through a threshold of $0.1$, then compute the incorrect mixing metrics.

### B.1    Simulations with nonlinear latent causal models and sensitivity to noisy scores

In Section 5, we have adopted linear Gaussian SEMs as latent causal models for which the score estimation can be done via estimating the precision matrices of the observed variables. In this section, we perform additional simulations to study nonlinear latent causal models to investigate the relation between the algorithm's performance and the accuracy of the score function estimates. To this end, we adopt a quadratic latent causal model under varying amounts of noise in score functions. Specifically, we follow the experimental setup in [7] and adopt a quadratic latent causal model with additive noise as follows

$$Z_i = \sqrt{Z_{\mathrm{pa}(i)}^\top \cdot \mathbf{A}_i \cdot Z_{\mathrm{pa}(i)}} + N_i , \tag{155}$$

where $\{\mathbf{A}_i : i \in [n]\}$ are positive-definite matrices, and the noise terms are zero-mean Gaussian variables with variances $\sigma_i^2$ sampled randomly from $\mathrm{Unif}([0.5, 1.5])$. For an intervention on node $i$, $Z_i$ is set to $N_i/2$. This causal model admits a closed-form score function (see [7, Appendix E.2] for

details), which enables us to obtain a score oracle. In our MN intervention setup, we use this score oracle and introduce varying levels of artificial noise according to

$$\hat{s}_X(x; \sigma^2) = s_X(x) \cdot \left(1 + \Xi\right), \quad \text{where} \quad \Xi \sim \mathcal{N}(0, \sigma^2 \cdot \mathbf{I}_{d \times d}) \tag{156}$$

to test the behavior of our algorithm under different noise regimes $\sigma \in [10^{-3}, 10^{-1.5}]$. Figure 1 demonstrates the results for the latent graph recovery and latent variable recovery.

