# OpenReview forum: "Linear Causal Representation Learning from Unknown Multi-node Interventions"
_NeurIPS.cc/2024/Conference — NeurIPS 2024 poster_

### Official Review · Reviewer_8PAL · 2024-07-04

**Soundness:** 3
**Presentation:** 4
**Contribution:** 3
**Rating:** 7
**Confidence:** 3

**Summary:**

This paper studies identifiability under unknown muilti-node  interventions (soft/hard), with general models (parametrtic/nonparametric) and **linear** mixing functions. This work provides both detailed proof which justifies the main theoretical statement, and a step-by-step algorithm which guides how to achieve identifiability in practice.
Overall, I find this work serves as an important step for interventional CRL towards more realistic settings.

&nbsp;

### References

[1] Burak Varıcı, Emre Acartürk, Karthikeyan Shanmugam, Abhishek Kumar, and Ali Tajer. Score- based causal representation learning with interventions. arXiv:2301.08230, 2023.

[2] Burak Varıcı, Emre Acartürk, Karthikeyan Shanmugam, Abhishek Kumar, and Ali Tajer. Score- based causal representation learning: Linear and general transformations. arXiv:2402.00849, 2024.

[3] Julius von Kügelgen, Michel Besserve, Wendong Liang, Luigi Gresele, Armin Kekic ́, Elias Bareinboim, David M Blei, and Bernhard Schölkopf. Nonparametric identifiability of causal rep- resentations from unknown interventions. In Proc. Advances in Neural Information Processing Systems, New Orleans, LA, December 2023.

**Strengths:**

This paper is extremely well written and clearly structured: it communicates clearly motivations, formulation, technical details, and theoretical implications. The experimental results adequately validate the theory in case of a linear causal model.

**Weaknesses:**

1. The proposed UMNI-CRL algorithm is claimed to work with *general* non-parametric causal models; however, the simulation experiment only showed results on *linear* structural equation model.  It would be great if the authors could report further experimental results on non-parametric causal models, to align with the theoretical claims. If there is a valid reason why it cannot be done, I am also very happy to hear.

2. Following the previous point, since this approach requires density estimation, it might not be scalable on nonparametric models. But to be fair, this seems to be a common limitation in many interventional CRL works [1, 2, 3].

3. Linearity assumption on the mixing function is restrictive, but the authors have acknowledged it and discussed possible future directions to overcome this limitation (sec. 6).

**Questions:**

See the first point in **weakness** section. I am very happy to raise my rating if this issue is resolved.

**Limitations:**

The authors discussed the remaining open problems and limitations in Section 6.

---

> ### Author Rebuttal · Authors · 2024-08-06
>
> We thank the reviewer for finding our results an important step towards more realistic CRL settings and noting the clarity of the paper.
>
> **General causal models**:
> We did not provide results using non-linear causal models since our algorithm, due to its combinatorial nature, is sensitive to input noise. While we are actively trying to make our algorithms work better using generic score estimators, for this response, in the interest of time, we have elected to provide a complementary analysis, where we investigate the performance of our algorithms under different levels of artificially introduced noise.
>
> For this set of experiments, we adopt a non-linear additive noise model with a score oracle. Specifically, the observational mechanism for node $i$ is generated according to
>
> \begin{align}
> Z_i = \sqrt{Z\_{{\rm pa}(i)}^\top {\bf A}_i Z\_{{\rm pa}(i)}} + N_i \ ,
> \end{align}
>
> where $N_i \sim {\cal N}(0, \sigma_i)$, and the interventional mechanisms set $Z_i = N_i / 2$. This causal model admits a closed-form score function (see [7, Eq.(393–395]), which enables us to obtain a score oracle. In our experiments, we use this score oracle and introduce varying levels of artificial noise according to
>
> \begin{align}
> \hat{s}_X(x; \sigma^2) = s_X(x) \cdot \big( 1 + \Xi \big) \ , \quad \mbox{where} \quad \Xi \sim {\cal N}(0, \sigma^2 \cdot {\bf I}\_{d \times d}) \ ,
> \end{align}
>
> to test the behavior of our algorithm under different noise regimes ($\sigma \in [10^{-3}, 10^{-1.5}]$). Results à la Table 2 versus different $\sigma$ values are provided in Figure 1 in the PDF document attached to the general response.
>
> **Score estimation**: We kindly note three things regarding the score functions on nonparametric models.
>
> - Our algorithm is agnostic to the choice of score estimator, and we can adopt popular score-matching methods as mentioned in Line 325, e.g., Song et al. (2020) or Zhou et al. (2020). We also note that score function estimation is finding increasingly more applications in diffusion models, and it is an active research field. Hence, our algorithms can modularly adopt any new score estimation algorithm and benefit from advances in the score estimation literature.
>
> - *Score vs. density estimation*: We also note that score estimation in a high-dimensional setting is generally easier than density estimation, as it avoids the difficulty of finding the normalizing constants.
>
>
> - Finally, we emphasize that our algorithm only requires **score differences** and does not require the score functions themselves. We conjecture that the direct estimation of score differences can be much more efficient than estimating the individual score functions, a lá direct estimation of precision difference matrices [R1]. Furthermore, density ratio estimation methods, e.g., classification-based approaches [R2], can be potentially useful for direct score difference estimation.
>
> [R1] Jiang, Wang, and Leng. "A direct approach for sparse quadratic discriminant analysis." JMLR, 2018.
>
> [R2] Gutmann and Aapo Hyvärinen. "Noise-contrastive estimation of unnormalized statistical models, with applications to natural image statistics." JMLR, 2012.

---

> > ### Comment · Reviewer_8PAL · 2024-08-08
> >
> > I thank the authors for clarifying and providing additional experiment results. I increased the score correspondingly.

---

### Official Review · Reviewer_NxzW · 2024-07-11

**Soundness:** 3
**Presentation:** 3
**Contribution:** 3
**Rating:** 7
**Confidence:** 2

**Summary:**

This paper advances Causal Representation Learning (CRL) by addressing the challenge of using unknown multi-node (UMN) interventions to identify latent causal variables and their structures.  The authors develop a score-based CRL algorithm that leverages UMN interventions to guarantee identifiability of latent variables and their causal graphs under both hard and soft interventions, achieving perfect identifiability with hard interventions and identifiability up to ancestors with soft interventions. Their method outperforms existing single-node approaches by ensuring robust recovery of causal structures in more complex, multi-intervention environments.

**Strengths:**

* Extending the causal representation learning to unknown multi-node interventions

* Proofs are provided

* Pseudocode is provided

* Computational complexity is discussed

* Limitations are clearly stated

**Weaknesses:**

* The paper primarily focuses on causal models with linear transformations. This limits its applicability in many real scenarios

* The applicability of the assumptions in real scenarios was not discussed

* The method was not applied on real world-data

**Questions:**

* Can you please elaborate on the computational complexity and on why it is dominated by step 2?

* Can you please discuss the applicability of the assumptions in real scenarios?

* I think that adding some real world application can increase the impact of this paper. Is it possible to find such an application?

**Limitations:**

The paper acknowledges certain limitation. One notable limitation is the assumption of linear transformations in the causal models considered. This restricts the applicability to scenarios where causal relationships are adequately approximated by linear relationships. Additionally, while the paper addresses the challenge of UMN interventions, it acknowledges the complexity involved in identifying intervention targets in such settings, which can affect the ability to fully leverage the statistical diversity inherent in UMN interventions.

---

> ### Author Rebuttal · Authors · 2024-08-06
>
> We thank the reviewer for their thoughtful review and accurate summary. We address the questions as follows.
>
> **Computational complexity**: Stage 1 (score difference estimation) is only performed once before the main algorithm starts. Stage 4 (unmixing procedure for hard interventions) essentially works as a post-processing step that does not pose computational challenges. Below, we elaborate on the computational complexity of Stage 2 and Stage 3.
>
> -   *Stage 2*: We check dimension of $\mathcal{V}$ (line 6 of the algorithm) at worst $n \times (2 \kappa + 1)^n$ times in Stage 2 of the algorithm. Here, $\kappa$ denotes the maximum possible determinant of a matrix $\\{0,1\\}^{(n-1) \times (n-1)}$. In the proof of Lemma 2 in Appendix A.1, we discuss why this choice facilitates the identifiability guarantees. We also discuss how $\kappa$ grows with $n$ in Appendix A.8, and give special cases, such as sparse multi-node interventions, in which $\kappa$ is upper bounded by small numbers even for large $n$ values.
>
> -   *Stage 3*: Here, we check dimension of $\mathcal{V}$ (line 23 of the algorithm) at worst $\\|\mathbf{W}\_{:,j}\\|_1 \times \\|\mathbf{W}\_{:,t}\\|_1$ times for every $(t,j)$ in Stage 3 of the algorithm. Note that $\mathbf{W}$ is updated within the steps of Stage 3. Therefore, the exact computational complexity would be a function of the graph structure, and the outputs of Stage 2.
>
> -   *Empirical tricks*: Finally, we note two empirical tricks that can reduce the computational complexity greatly. First, even though $\kappa$ can grow quickly as $n$ becomes larger, in practice, setting $\kappa=2$ usually works fine (see additional experiment results in the general response). Second, additional empirical tricks can greatly increase the speed of the algorithm, e.g., dividing the columns of $\mathbf{W}$ by the greatest common divisor of its entries after every step. Since our focus is on establishing the identifiability results, we omit the investigation of empirical tricks that would disturb the flow of the paper and distract from the main focus.
>
>
> **Toward realistic settings**: We believe this paper serves as a significant step in this direction by removing the stringent assumption of single-node interventions. For instance, genomics datasets such as Perturb-seq (Norman et al. 2019) are used by single-node interventional CRL (Zhang et al. 2023). However, the genomic interventions are known to have unknown off-target effects (Fu et al. 2013, Squires et al. 2020) that violate the single-node intervention assumption. Therefore, establishing unknown multi-node interventional results is fundamental to unlocking the use of CRL in realistic datasets. The implementation of real applications is beyond the scope of this work and is an important future direction.
>
> **References**
>
> T. M. Norman, M. A. Horlbeck, J. M. Replogle, A. Y. Ge, A. Xu, M. Jost, L. A. Gilbert, and J. S. Weissman. Exploring genetic interaction manifolds constructed from rich single-cell phenotypes. Science, 365(6455):786–793, 2019.
>
> J. Zhang, C. Squires, K. Greenewald, A. Srivastava, K. Shanmugam, and C. Uhler. Identifiability guarantees for causal disentanglement from soft interventions. NeurIPS 2023
>
> Y. Fu, J. A. Foden, C. Khayter, M. L. Maeder, D. Reyon, J. K. Joung, and J. D. Sander, “High frequency off-target mutagenesis induced by CRISPR-Cas nucleases in human cells,” Nature Biotechnology, vol. 31, no. 9, pp. 822–826, 2013
>
> C. Squires, Y. Wang, and C. Uhler. Permutation-based causal structure learning with unknown intervention targets. UAI 2020

---

> > ### Comment · Reviewer_NxzW · 2024-08-11
> >
> > I thank the authors for the response. After reviewing the reviews and considering the responses, I will raise my score to 7.

---

### Official Review · Reviewer_Vtz8 · 2024-07-12

**Soundness:** 4
**Presentation:** 3
**Contribution:** 3
**Rating:** 8
**Confidence:** 4

**Summary:**

This work studies interventional causal representation learning, where one has access to interventional data, to identify latent causal factors and latent DAG in the unknown multi-node interventions regime. The authors consider a setting where the mixing function is linear and the latent causal model is nonparametric. Under the assumption of sufficient interventional diversity, the authors use score function arguments to show that the underlying causal factors of variation (and DAG) can be recovered (1) up to permutation and scaling from stochastic hard interventions and (2) up to ancestors from soft interventions. The authors propose a score-based framework (UMNI-CRL) and evaluate it on synthetic data generated from Erdős–Rényi random graph model.

**Strengths:**

- This work provides significant results in the unknown multi-node intervention setting, which is much more realistic than the common single-node intervention regime. As opposed to other works, this work studies CRL from a more general class of multi-node interventions (stochastic hard and soft).
- The paper is well-written, the concepts are explained well, and the theoretical identifiability results add a lot of value to the current CRL literature.
- The use of score functions and score differences in the observation space to estimate the unmixing function, especially for the UMN setting, is a novel and interesting approach for CRL.
- This work is the first to establish latent DAG recovery in the UMN setting under any type of multi-node intervention for arbitrary nonparametric latent causal models.

**Weaknesses:**

Although the theoretical contribution of this work is strong, the empirical evaluation is quite weak compared to other works in CRL. There are only experiments for n=4 causal variables. There is also no baseline comparison of the proposed framework with other methods in the UMN setting (e.g., [1]). Also, some discussions are a bit abridged and could use more elaboration in the paper (see below for details).

[1] Bing et al. “Identifying Linearly-Mixed Causal Representations from Multi-Node Interventions” CLeaR 2024.

**Questions:**

- I would like some clarification on the intervention regularity condition. Specifically, why does the additional term ensure that multi-node interventions have a different effect on different nodes? It would be good to elaborate on this condition when introduced since it is a central assumption that needs to be satisfied for the results to hold.
- How do you obtain $\Lambda$ in Eq. (14)? It seems that this matrix encodes the summands with the latent space score differences. However, since the distribution of the latents is unknown, how would you go about estimating $\Lambda$ and score differences $\Delta S_X$ in general cases of nonparametric distributions?
- How do you learn the integer-valued vectors $\mathbf{w}$ in Stage 2 of the algorithm? From Eq (18), it seems that $\mathcal{W}$ is a fixed predefined set and you choose the vectors $\mathbf{w} \in \mathcal{W}$ that satisfy a specific condition in the algorithm. To my understanding, this is central to recovering the approximate unmixing $\mathbf{H}^*$ up to a combination of the rows of the true unmixing $\mathbf{G}^{\dagger}$. I would appreciate it if the authors could elaborate on how this procedure was done.
- From Appendix A.8, it seems that $\kappa$ is determined by the number of causal variables $n$. Could the authors give some more intuition on what $\kappa$ represents in Stage 2 with respect to how the unmixing is recovered?
- Are there any distributional assumptions on the exogenous noise in the latent additive noise causal model?
- It seems that the UMN hard intervention result (Theorem 1) requires a latent model with additive noise. Would perfect recovery still be possible for latent models with non-additive noise under UMN hard interventions?
- The empirical results suggest that increasing sample size improves DAG recovery, which is intuitive. However, what do the results look like as the number of causal variables scales up? Currently, the authors only show results for n=4 latent causal variables. I only offer this as a suggestion due to the short rebuttal period.
- How would the assumptions made need to change to be applied to general mixing functions? I know that generality in one aspect of the model (i.e., general SCM) may require other aspects to take some parametric form (i.e., linear mixing) for identifiability guarantees, but do the authors have any intuition on how to achieve identifiability results for the UMN setting in a completely nonparametric setup?

**Limitations:**

Limitations are discussed in Section 6.

---

> ### Author Rebuttal · Authors · 2024-08-06
>
> We thank the reviewer for acknowledging our strong theoretical and algorithmic contributions. We address the questions as follows.
>
> **Empirical results**:
> -   *Increasing $n$*: Thanks for the suggestion. Please refer to the general response for experiment results for up to $n=8$ nodes.
> -   *Baseline*: We note that Bing et al. use “do” interventions. Hence, the algorithms are not comparable.
>
> **Interventional regularity**: We explain the rationale and details as follows.
>
> - First, we emphasize that **interventional regularity is not a restrictive assumption**. In Appendix A.8.(Lemma 5), we present some sufficient conditions in which the interventional regularity holds. For instance, under a hard intervention on a linear causal model, if the distribution of the exogenous noise remains the same, then interventional regularity holds. Hence, it is not a restrictive assumption.
>
> - Next, we clarify what we mean by the *effect of an intervention*. Essentially, $\frac{\partial\log p_i/q_i}{\partial z_j}$ is the effect of intervening on node $i$ on the *score* associated with node $j$. Note that we use *combinations* of different multi-node environments to generate new score difference functions. Without the additional term in Eq.(12), $\frac{\partial\log p_j/q_j}{\partial z_j}$, the ratio in Eq.(11) considers only a single node intervention on $i$ and its effects on scores of $i$ and $j$.
>
> - To ensure that the effect of a *combined* intervention is different on scores of $i$ and $j$, we need to consider interventions on both $i$ and $j$. As such, the additional term in Eq.(12), $\frac{\partial\log p_j/q_j}{\partial z_j}$, denotes the effect of the additional intervention on $j$ on the node $j$.
>
> **Score function differences**:
> -   **We do not require estimating $\Lambda$**. It is correct that $\Lambda$ encodes score differences in latent space, and we cannot estimate it. Therefore, the algorithm **only** takes $\Delta S_X$ as input. $\Lambda$ is defined to provide intuition on the rationale of the algorithm and the connection between latent score differences and $\Delta S_X$ in Eq.(17).
>
> -   Our algorithm is agnostic to the choice of score estimator, and we can adopt popular score-matching methods as mentioned in Line 325, e.g., Song et al. (2020) or Zhou et al. (2020). We also note that score function estimation is finding increasingly more applications in diffusion models, and it is an active research field. Hence, our algorithms can modularly adopt any new score estimation algorithm and benefit from advances in the score estimation literature.
>
> -   We also emphasize that our algorithm only requires **score differences** and does not require the score functions themselves. We expect that the direct estimation of score differences can be much more efficient than estimating the individual score functions, à la direct estimation of precision difference matrices [R1]. Furthermore, density ratio estimation methods, e.g., classification-based approaches [R2], can be potentially useful for direct score difference estimation.
>
> [R1] Jiang et al. A direct approach for sparse quadratic discriminant analysis. JMLR, 2018.
>
> [R2] Gutmann and Hyvärinen. Noise-contrastive estimation of unnormalized statistical models, with applications to natural image statistics. JMLR, 2012.
>
> **Additive noise models (ANM)**: Please refer to our general response.
>
> **Stage 2 of the algorithm**: We elaborate on learning the vectors $\bf w$ and the role of $\kappa$ as follows.
> -   Note that $\Delta S_X\cdot\bf w$ is essentially a combination of SN latent score differences via Eq.(15) and (17) (also shown in (30)). Also, since the score difference $\nabla\log p_i(z_i\mid z_{{\rm pa}(i)})-\nabla\log q_i(z_i\mid z_{{\rm pa}(i)})$ is a function of $Z_{{\rm pa}(i)}$ and $Z_i$, the dimension of the image of this function will be 1 if and only if the intervened node $i$ has no parents.
>
> -   Leveraging this property, at the first step $t=1$, we search for a linear combination of the given MN environments (via $\bf w$) to emulate a single node intervention on a root node. For such a vector $\bf w$, using Eq.(15) and (17), the image of $\Delta S_X\cdot\bf w$ contains only one vector (up to scaling), which is the encoder $\bf G^\dagger$’s row corresponding to the $i$-th node where $i$ is a root node.
>
> -   Subsequently, at each step, we follow the same routine to estimate a row of the true encoder. While checking the dimension of the emulated intervention, we project $\Delta S_X\cdot\bf w$ to the nullspace of the submatrix recovered so far. This ensures that the learned encoder will be full-rank.
>
> -   **Role of $\kappa$**: Vector ${\bf w=[D^{-1}]}_i$ is a valid choice that makes $\Delta S_X\cdot\bf w$ correspond to a single node intervention. Then, by constructing a set $\cal W$ that contains the rows of $\bf D^{-1}$, we ensure that the procedure described above will work by an exhaustive search over $\cal W$. We note that the entries of $\bf D^{-1}$ can be found via the cofactor matrix of $\bf D$, for which the $\kappa$ denotes the maximum possible entry. For detailed derivation, please refer to lines 479-490 in Appendix A.1.
>
> **Intuition for general mixing functions**: Our core technical idea is using combinations of MN interventions to construct new interventions with desired properties, e.g.,sparsity. Our intuition is that we can extend the intervention matrix in Eq.(4) to handle multiple interventional mechanisms. For instance, to represent two interventional mechanisms per node, the columns of the intervention matrix would be in $\\{0,1\\}^{2n}$. Subsequently, the goal is to find combinations of MN environments to create two different SN interventions for each node so that the problem simplifies to the known results for general transformations under two SN int/node (see references [6],[9]). However, building on this intuition to prove identifiability is challenging and is a major direction for future work.

---

> > ### Comment · Reviewer_Vtz8 · 2024-08-09
> >
> > I greatly appreciate the authors taking the time to answer my questions and provide clarifications. My questions and concerns have been addressed quite well in the response. The new empirical results for a larger number of causal variables further strengthen the paper. I believe this is a high-quality submission with significant theoretical results of great interest to the CRL community. Thus, I raise my score to 8.

---

### Official Review · Reviewer_q4pN · 2024-07-12

**Soundness:** 3
**Presentation:** 3
**Contribution:** 2
**Rating:** 6
**Confidence:** 4

**Summary:**

This paper extends previous results on using score function for causal representation learning to the settings with unknown multi-node interventions. This new setting poses significant new challenges as opposed to the single node intervention case. The author first present theoretical identifiability result on hard interventions with latent additive noise model and on soft interventions. They then propose an algorithm called (UMNI)-CRL and test it on synthetic linear Gaussian dataset.

**Strengths:**

The paper is clearly written, easy to follow and with good motivations.

**Weaknesses:**

1. The transformation from latent to observed is noiseless, which could be a limitation.
2. Line 199 says that: “This regularity condition ensures that the effect of a multi-node intervention is not the same on different nodes”. But how realistic or neccessary is this condition? It seems like it is very possible that an intervention can cause two downstream nodes to have the same effect although these two nodes is not influenced the same by all type of interventions.
3. The experiments are only on synthetic dataset but I don’t think that is a big issue.
4. Some potential missing citations

    [1] Kumar, Abhinav, and Gaurav Sinha. "Disentangling mixtures of unknown causal interventions." *Uncertainty in Artificial Intelligence*. PMLR, 2021.

    [2] Jiang, Yibo, and Bryon Aragam. "Learning nonparametric latent causal graphs with unknown interventions." *Advances in Neural Information Processing Systems* 36 (2024).

**Questions:**

1. (UMNI)-CRL requires estimating the score function. How do you ensure a good estimate of the score function to unsure that the algorithm is useful in practice?
2. One small question: on line 141-143, it is mentioned that if a node is not intervened on, perfect identifiability is not possible. But there are cases like A→B where I don’t need to intervene on A?

**Limitations:**

1. Theorem 1 only works for additive noise model.
2. The transformation from latent to observed noiseless.
3. Experiments are only on synthetic dataset.
4. I am unsure if the algorithm is practical because it needs to estimate the score function.

---

> ### Author Rebuttal · Authors · 2024-08-06
>
> We thank the reviewer for acknowledging the challenges of the unknown multi-node intervention setting and noting the clarity of the paper. We address the raised concerns as follows.
>
> **Noiseless transformation:** The current scope of the paper cannot handle noisy transformations. We also kindly note that the closely related CRL literature (see references [3]-[10] in the paper) also considers deterministic transformations $X = g(Z)$. Our paper’s primary goal is to relax the restrictive assumption of single-node interventions. As such, we consider noisy transformations a major future direction.
>
> **Interventional regularity**: We thank the reviewer for raising the need for elaboration on the interventional regularity condition. We explain its rationale and emphasize that it is not a restrictive assumption as follows.
>
> - First, we want to clarify what we meant by "effect of an intervention.” Essentially, $\frac{\partial}{\partial z_j}\log\frac{p_i(z_i\mid z_{{\rm pa}(i)})}{q_i(z_i\mid z_{{\rm pa}(i)})}$ is the effect of intervening on node $i$ on the *score* associated with node $j$. Therefore, the effect in the score function is not on downstream nodes of $i$ but on the parents of $i$.
>
> - Note that we use *combinations* of different multi-node environments to generate new score difference functions. Therefore, to ensure that the effect of this new “combined” intervention will be different on the $i$-th and $j$-th coordinates of the score function, we require the ratio in Eq.(12) to be not constant.
>
> - Next, we emphasize that **interventional regularity is not a restrictive assumption**. In Appendix A.8., we present some sufficient conditions that make the interventional regularity valid (see Lemma 5). For instance, if we consider a linear latent causal model under a hard intervention and the distribution of the exogenous noise term remains the same after the intervention, then interventional regularity is satisfied. Therefore, interventional regularity is not a restrictive assumption.
>
> - Furthermore, note that if there exist $k\in{\rm pa}(j)\setminus{\rm pa}(i)$, then $\log\frac{p_j(z_j\mid z_{{\rm pa}(j)})}{q_j(z_j\mid z_{{\rm pa}(j)})}$ is a function of $Z_k$, whereas the other terms in Eq.(12) are not. This implies that the ratio in Eq.(12) is not a constant function of $Z$ and again exemplifies that the interventional regularity is not a restrictive condition.
>
> **Estimating score function differences:**
> -    Our algorithm is agnostic to the choice of score estimator, and we can adopt popular score-matching methods as mentioned in Line 325, e.g., Song et al. (2020) or Zhou et al. (2020). We also note that score function estimation is finding increasingly more applications in diffusion models, and it is an active research field. Hence, our algorithms can modularly adopt any new score estimation algorithm and benefit from advances in the score estimation literature.
>
> -    We also emphasize that our algorithm only requires **score differences** and does not require the score functions themselves. We conjecture that the direct estimation of score differences can be much more efficient than estimating the individual score functions, a lá direct estimation of precision difference matrices [R1]. Furthermore, density ratio estimation methods, e.g., classification-based approaches [R2], can be potentially useful for direct score difference estimation.
>
> [R1] Jiang, Wang, and Leng. "A direct approach for sparse quadratic discriminant analysis." JMLR, 2018.
>
> [R2] Gutmann and Aapo Hyvärinen. "Noise-contrastive estimation of unnormalized statistical models, with applications to natural image statistics." JMLR, 2012.
>
> **Non-identifiability when missing an intervened node**: In lines 141-143, we meant that “when a node is not intervened on, perfect identifiability is not possible **in general**”, i.e., without imposing additional restrictions such as structural assumptions. Our reference for non-identifiability is Proposition 5 of Squires et al. (2023). Their proof for non-identifiability only requires that the non-intervened node $i$ has at least one parent. We will add this note to the updated manuscript.
>
> We note that the example of $A\rightarrow B$ under an intervention on only $B$ is also discussed by Remark 2 of Squires et al. (2023). Even though the graph $A\rightarrow B$ can be discovered in this specific case, we are not aware of any results for the identifiability of the latent variables. Squires et al. (2023) empirically suggest that the latent variables are not identifiable.
>
> **Additive noise models (ANM)**: Please refer to our general response.
>
> **Additional references**: Thank you for the suggestions; we will include and discuss them in the paper. In summary, Jiang and Aragam (NeurIPS 2023) focus on recovering the latent DAG without recovering the latent variables as opposed to our complete CRL objectives. On the other hand, Kumar and Sinha (2021) focus on an entirely different setting in which the causal variables are observed, and the distributions are given in a mixture. In contrast, our CRL setting focuses on latent variables observed through the same transformation when observing distinct interventional distributions.

---

> > ### Comment · Reviewer_q4pN · 2024-08-12
> >
> > Thanks for your rebuttal. I have raised my score leaning towards acceptance.

---

### Official Review · Reviewer_GLEY · 2024-07-15

**Soundness:** 3
**Presentation:** 3
**Contribution:** 3
**Rating:** 7
**Confidence:** 3

**Summary:**

This paper introduces new identifiability results for CRL in environments with unknown multi-node interventions. It shows that, with sufficiently diverse interventional environments, one can achieve identifiability up to ancestors using soft interventions and perfect identifiability using hard interventions. The paper also provides an algorithm with identifiability guarantees.

**Strengths:**

- The paper tackles the complex and underexplored multi-node intervention setting. The established identifiability can be crucial for extending current CRL theories into more practical contexts.
- The introduced algorithm that leverages score functions with different interventional environments is also interesting and insightful.
- The paper is well-motivated and articulated with high clarity.

**Weaknesses:**

- The proposed algorithm, while theoretically sound, seems computationally demanding. In fact, even a 4-node low-dimensional case requires a large number of environments and samples. The paper could benefit from a deeper discussion on the scalability of the algorithm.
- The current evaluation of the algorithm is limited to synthetic simulations. Expanding it to more realistic datasets would substantively improve its practical significance.

**Questions:**

How effectively does the proposed algorithm scale to more nodes and higher dimensions?

**Limitations:**

The paper acknowledges its main limitations in the reliance on linear transformations.

---

> ### Author Rebuttal · Authors · 2024-08-06
>
> We thank the reviewer for finding our results crucial for extending CRL into more practical contexts, for finding our algorithm insightful, and for noting the clarity of the paper. We address the raised questions about the algorithm’s scalability as follows.
>
> -    **Dimension of $X$**: Our algorithm is readily scalable to arbitrarily high-dimensional observations $X$. Please see the additional experiments reported in our general response, in which we use $d=50$.
>
> -    **Dimension of $Z$**: Please see the additional experiments reported in our general response, in which we use $n \in \\{4, 5, 6, 7, 8\\}$ latent nodes.
>
> -    **Number of environments**: We note that $n$ environments are *necessary* in general (without further structural assumptions) for identifiability via single-node interventions (shown by Squires et al. 2023, Proposition 5). Since our unknown multi-node intervention setting subsumes the single-node interventions, we require at least $n$ environments as well.
>
> -    **Number of samples**: We remark that the main purpose of our algorithm is to establish a framework for identifiability via multi-node interventions. In future work, we aim to address the efficient score difference estimation, which is a separate line of work that can significantly increase the efficiency of our framework. We also kindly note that, even in the much simpler single-node intervention setting, related CRL literature uses a similar number of samples to $10^5$ that we used in our experiments for good performance (e.g., $10^5$ samples for $n=5$ nodes in Squires et al. (2023), $5 \times 10^4$ samples for $n=5$ nodes in Buchholz et al. (2023), and  $5 \times 10^4$ samples for $n=5$ nodes in Varici et al. (2024)).
>
> -    **Toward realistic settings**: Finally, we acknowledge the need for working with more realistic datasets in the CRL field. We believe this paper serves as a significant step in this direction by removing the stringent assumption of single-node interventions, and we leave addressing realistic applications to future work.

---

> > ### Comment · Reviewer_GLEY · 2024-08-13
> >
> > Thank you for your response to my question. I increased the score to 7.

---

### Author Rebuttal · Authors · 2024-08-06

We thank all the reviewers for their thorough feedback and thoughtful questions. Below we address some shared questions by the reviewers.

### **Additional experiments**
We address the shared concerns of the reviewers regarding the scalability of the algorithm via the following additional experiment results.

**Setting**: We follow the same setting as in Section 5 of the paper; Erdös-Rényi model with density 0.5 and linear structural equation models (SEMs) with Gaussian noise.
- **Dimension of observed variables $X$**: We increase $d$ to $50$ in this additional experiments.
- **Number of latent nodes**: We perform experiments for $n \in \\{4,5,6,7,8\\}$ nodes. We also note that $n=8$ is the largest graph size considered in the closely related single-node interventional CRL literature (e.g., Squires et al. (2023) and Buchholz et al. (2023) consider 5 nodes, Varici et al. (2024) consider 8 nodes, von Kügelgen et al. (2023) consider 2 nodes).
- We use $n_{\rm s} = 10^5$ samples for each realization and repeat the experiments 100 times for each $(n,d)$ pair.

The table below shows that the average rate of incorrect mixing entries, captured by $\ell_{\rm soft}$ and $\ell_{\rm hard}$, remains low for increasing values of $n$. Graph recovery metric SHD increases with $n$, since the number of expected edges also increases with $n$ under a fixed edge density.

|    $n$   |    $d$    |  SHD (Soft)  | $\ell_{\rm soft}$ |  SHD (Hard) | $\ell_{\rm hard}$ |
|:--------:|:---------:|:------------:|:-----------------:|:-----------:|:-----------------:|
|     4    |     50    |      0.44    |        0.72       |     0.04    |        0.11       |
|     5    |     50    |      0.96    |        1.25       |     0.05    |        0.10       |
|     6    |     50    |      2.41    |        3.20       |     0.09    |        0.14       |
|     7    |     50    |      4.22    |        6.00       |     0.11    |        0.16       |
|     8    |     50    |      5.67    |        8.75       |     0.10    |        0.16       |

**Empirical trick**: We recall that our algorithm involves searching for proper $\mathbf{w} \in \\{-\kappa,\dots,\kappa\\}^n$ vectors in which $\kappa$ denotes $\kappa$ denotes the maximum determinant of a matrix in $\\{0,1\\}^{(n-1) \times (n-1)}$. Even though $\kappa$ is a function of $n$, e.g., $\kappa=2$ for $n=4$ and $\kappa=5$ for $n=6$, we observe that setting $\kappa=2$ does not disturb the performance noticeably. Therefore, we set $\kappa=2$ in all our experiments to reduce runtime.

### **General causal models**
In addition to our experiments with linear causal models, we investigate general causal models. Specifically, we provide a sensitivity analysis where we investigate the performance of our algorithms under different levels of artificially introduced noise.

For this set of experiments, we adopt a non-linear additive noise model with a score oracle. Specifically, the observational mechanism for node $i$ is generated according to
$$Z_i = \sqrt{Z\_{{\rm pa}(i)}^\top {\bf A}_i Z\_{{\rm pa}(i)}} + N_i \ ,$$
where $N_i \sim {\cal N}(0, \sigma_i)$, and the interventional mechanisms set $Z_i = N_i / 2$. This causal model admits a closed-form score function (see [7, Eq.(393–395]), which enables us to obtain a score oracle. In our experiments, we use this score oracle and introduce varying levels of artificial noise according to
$$\hat{s}_X(x; \sigma^2) = s_X(x) \cdot \big( 1 + \Xi \big) \ , \quad \mbox{where} \quad \Xi \sim {\cal N}(0, \sigma^2 \cdot {\bf I}\_{d \times d}) \ ,$$
to test the behavior of our algorithm under different noise regimes ($\sigma \in [10^{-3}, 10^{-1.5}]$). Results à la Table 2 versus different $\sigma$ values are provided in Figure 1 in the PDF document attached to the general response.


### **Additive noise models (ANM)**
 -   We emphasize that the core component of our work – minimizing score differences to estimate the true encoder – does not require ANMs, as shown by Theorem 2 for soft interventions.
-   ANM is introduced for the analysis of CI tests in Stage 4 (hard interventions). Specifically, given a different causal model, it may be possible to analyze Stage 4 of the algorithm in a different way. For simplicity, we have adopted ANMs which are commonly adopted by both causal discovery and CRL literature (e.g., for perfect identifiability, Squires et al. (2023), Buchholz et al. (2023), Varici et al. (2024), and Bing et al. (2024) use ANMs).
-   Finally, we only require the exogenous noise in the ANM to have full support, which is already implied by the full support of $z$. Hence, we don’t make any specific assumptions for ANM.

---

### Decision · Program_Chairs · 2024-09-25

**Decision:**

Accept (poster)

**Comment:**

This paper develops new results for multi-node causal representation learning, which is an important open direction for the field. All reviewers are in favor of acceptance. Clear accept.